# Lysine myristoylation mediates long-term potentiation via membrane enrichment of synaptic plasticity effectors

Benjamin Matthews[1], Sevannah A Steeves[1], Isaac O Akefe[1,4], Noorya Yasmin Ahmed[1,2], Rachel S Gormal [1], Nathalie Dehorter [1,2], Tristan P Wallis [1]✉ & Frédéric A Meunier [1,3]✉

## Abstract

Synaptic plasticity underlying long-term memory is associated with the generation of saturated free fatty acids (sFFAs) –particularly myristic acid– from membrane phospholipids by the phospholipase A1 isoform DDHD2. However, the mechanism through which myristic acid contributes to synaptic plasticity remains elusive. Here we demonstrate that DDHD2-derived myristic acid is rapidly converted to myristoyl CoA, which serves as the substrate for N-myristoyl transferases (NMT1/2), to promote post-translational lysine myristoylation of synaptic proteins. Chemically-induced long-term potentiation (cLTP) in cortical neurons increases both sFFAs and their CoA-conjugates, predominantly myristoyl CoA, and this response is blocked by the DDHD2 inhibitor KLH-45. KLH-45-mediated inhibition of DDHD2 or IMP-1088-mediated inhibition of NMT1/2 also disrupts cLTP-induced proteomic changes, impairs dendritic spine remodeling, and prevents LTP in hippocampal slices. Instrumental conditioning further induces proteomic changes in the hippocampus, which are abolished in learning-deficient DDHD2$^{-/-}$ knockout mice. In these mice, key synaptic proteins such as NMDA receptor subunit GluN1, MAP2, and GAS7 fail to undergo learning-induced changes, effectively linking DDHD2 function to learning-dependent proteome remodeling. Our findings reveal that de novo lysine myristoylation promotes synaptic plasticity and memory formation.

**Keywords** DDHD2; *N*-Myristoyl Transferase; Myristic Acid; Lysine Myristoylation; Long-Term Potentiation
**Subject Categories** Neuroscience; Post-translational Modifications & Proteolysis

## Introduction

The neuronal lipidome undergoes dynamic change during memory and learning, and perturbations of these responses are a key hallmark of many neurodegenerative diseases (Bonelli et al, 2020; Estes et al, 2021; Singh et al, 2023). This is exemplified by the recent discovery that the activity of the phospholipase A1 enzyme DDHD2 is required for memory acquisition. DDHD2 is targeted to the synaptic membrane via binding to STXBP1, where it hydrolyses phospholipids at their *sn-1* glycerol ester to release a saturated free fatty acid (sFFA) and the corresponding lysophospholipid during memory formation (Akefe et al, 2024). This process results in a dramatic relative increase in the sFFA myristic acid far in excess of the relative increases in more abundant sFFAs such as palmitic (C16:0) and stearic (C18:0) acids. The activity-dependent release of myristic acid has been shown to occur as a response to stimulation of cultured neurons and chromaffin cells in vitro (Narayana et al, 2015) and also increases in rat brain regions associated with memory acquisition during fear conditioning (Wallis et al, 2021) and instrumental learning in mice (Akefe et al, 2024). DDHD2 knock-out mice display learning and memory symptoms recapitulating the cognitive and neuromuscular deficits displayed in human spastic paraplegia (HSP) patients with DDHD2 variants (Inloes et al, 2014). Importantly, these mice are deficient in their activity-dependent myristic acid response (Akefe et al, 2024), suggesting that myristic acid is a major driver of synaptic plasticity.

How myristic acid contributes to these changes is underexplored. Protein lipidation mediates membrane localisation, proteolytic degradation and can stabilise protein complexes (Jiang et al, 2018). In particular, palmitoylation has been shown to affect synaptic plasticity (Brigidi et al, 2014). We hypothesise that myristic acid is used for myristoylation of synaptic proteins that underpin the plastic changes occurring during learning. Protein myristoylation occurs as an amide linkage of the myristoyl group to either the *N*-terminus (Yuan et al, 2020) or on lysine residues (Kosciuk and Lin, 2020). Both *N*-terminal and lysine myristoylation are catalysed by the same enzymes, *N*-myristoyl transferase 1 and 2 (NMT1/2), but differ markedly in that *N*-terminal myristoylation

[1]Clem Jones Centre for Ageing Dementia Research, Queensland Brain Institute, The University of Queensland, St Lucia, QLD 4072, Australia. [2]The John Curtin School of Medical Research, The Australian National University, Canberra, ACT, Australia. [3]The School of Biomedical Sciences, The University of Queensland, St Lucia, QLD 4072, Australia. [4]Present address: CDU Menzies School of Medicine, Charles Darwin University, Ellengowan Drive, Darwin, NT 0909, Australia. ✉E-mail: t.wallis@uq.edu.au; f.meunier@uq.edu.au

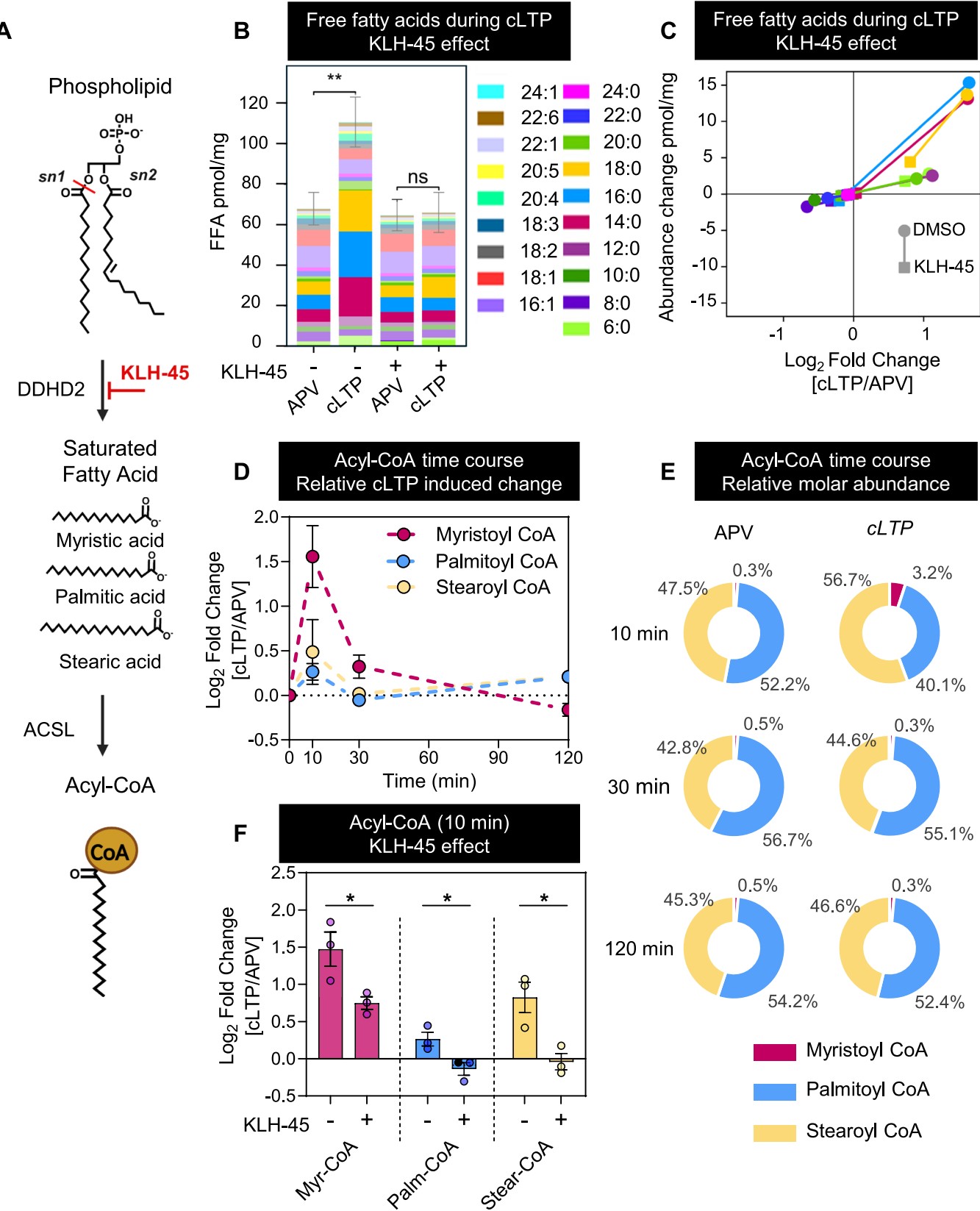

◄   **Figure 1.   DDHD2 regulates the cLTP-responsive generation of saturated free fatty acids (FFA) and their coenzyme A conjugates.**

(A) Schematic showing the generation of saturated free fatty acids by phospholipase A1 isoform DDHD2 through the cleavage of acyl chains at the sn-1 position of phospholipids. DDHD2 activity is specifically inhibited by KLH-45. Free fatty acids are converted to metabolically available forms through conjugation to coenzyme A by long-chain acyl CoA synthetases (ACSL). (B) Stacked bar plots showing the absolute changes in the 19 FFAs measured by FFAST LC–MS/MS analysis 10 min after glycine-induced cLTP in KLH-45 and DMSO (vehicle) treated cortical neurons. (C) Linked scatter plot showing cLTP-induced relative change (log$_2$ [cLTP/APV]) on the x-axis and absolute concentration change in pmol/mg tissue (cLTP-APV) on the y-axis for the sFFAs. The effect of DDHD2 inhibition is indicated by the connecting line between the DMSO vehicle (circle) and KLH-45 treatment (square) for each sFFA. (D) Time course analysis of the relative change in response to cLTP (Log$_2$ [cLTP conc/APV conc]) in myristoyl CoA, stearoyl CoA and palmitoyl CoA 10, 30 and 120 min after glycine stimulation. (E) Pie charts displaying the percentage relative molar concentration of myristoyl CoA, stearoyl CoA and palmitoyl CoA at 10, 30 and 120 min post-glycine stimulation in cLTP-treated samples and APV-treated controls (F) Bar plots indicating the relative fold change of the three acyl CoA species in response to cLTP (Log$_2$ [cLTP conc/APV conc]) 10 min after glycine stimulation in the presence of KLH-45. Data information: (B) FFA analysis was performed using three independent biological replicates. Statistical significance was determined using an unpaired Student's $t$-test with Holm–Sidak post hoc correction. cLTP-induced change corrected $p$ values: Vehicle control $p = 0.005$; KLH-45 treated $p = 0.832$. **$p < 0.01$, ns not significant. Error bars indicate the cumulative standard error of the mean (SEM) for the summed FFAs in each treatment. (D) Analysis was performed at three independent time points, each with three biological replicates. Error bars indicate SEM for each data point. Statistical significance was determined using repeated measures two-way ANOVA, with cumulative $p = 0.0005$, Acyl CoA species effect $p = 0.0372$ and a time effect $p = 0.0129$. (F) Acyl CoA analysis was performed using three independent biological replicates. Error bars indicate SEM. Statistical significance was measured using two-way ANOVA (Acyl CoA species effect $p = 0.0001$; Inhibitor treatment effect $p < 0.0001$). Pairwise $t$-test analysis of the difference between vehicle and KLH-45 treatment: Myristoyl CoA $p = 0.04$; Palmitoyl CoA $p = 0.03$; Stearoyl CoA $p = 0.02$. *$p < 0.05$.

(N-myristoylation) predominantly occurs co-translationally, while lysine myristoylation (K-myristoylation) is post-translational (Yuan et al, 2020). Co-translational myristoylation occurs at N-terminal glycine residues after removal of the initiator methionine (Thinon et al, 2014). As no known mammalian deacylase has been characterised that removes N-terminal myristoyl groups, it is likely that this modification persists for the lifetime of the protein. Since myristoylation was first characterised in 1982 (Aitken et al, 1982; Carr et al, 1982), an expanding list of myristoylated proteins have been described, with the Uniprot database containing 201 human and 177 mouse proteins annotated as N-terminally myristoylated. In contrast, post-translational lysine myristoylation occurs on mature proteins and can be readily removed through the activity of an expanding range of deacylases (Komaniecki and Lin, 2021). Tumour necrosis factor alpha was first shown to be post-translationally myristoylated, leading to inhibition of its secretion and ultimate lysosomal degradation (Stevenson et al, 1992). Recent work has shed more light on the functions of lysine myristoylation, including a role in regulating the GTP-dependent localisation of ARF6 (Kosciuk and Lin, 2020) and Ras (Zhang et al, 2017), and functional regulation of protein kinase A (Bagchi et al, 2022). These examples are, however, still limited, and the full scope of the biological role of this modification remains underexplored.

In this study, we demonstrate that synaptic plasticity is underpinned by DDHD2-driven generation of myristic acid, which is subsequently converted to myristoyl Coenzyme A (CoA) as a substrate for synaptic protein myristoylation via NMT1/2. We examined the lipidomic, proteomic and electrophysiological response to pharmacological and genetic manipulation of this pathway, to reveal that K-myristoylation rather than N-myristoylation underpins synaptic plasticity and learning.

# Results

## Myristic acid generated by DDHD2 is converted to myristoyl CoA during long-term potentiation

We recently demonstrated that sFFAs increase in response to neuronal stimulation in vitro (Narayana et al, 2015) and memory

acquisition in vivo (Wallis et al, 2021) via the action of the PLA1 isoform DDHD2 (Akefe et al, 2024). The main sFFAs generated were myristic, palmitic and stearic acids (Akefe et al, 2024), raising the possibility that these could be used for lipidation of synaptic proteins as an underlying mechanism enabling synaptic plasticity. To be used as substrate to lipidate synaptic proteins, sFFAs need to be conjugated to CoA (Fig. 1A). We induced synaptic plasticity in cultured mouse cortical neurons using a chemically induced long-term potentiation (cLTP) protocol. This involved stimulation with glycine to allosterically activate post-synaptic N-methyl-D-aspartate receptors (NMDAR) (Zhang et al, 2020). To ensure that the measured responses were due to NMDAR activation, cultures were treated with a suite of neurotoxins to inhibit off-target glycine signalling. This included treatment with strychnine and bicuculline as competitive glycine antagonists to abrogate activation of inhibitory post-synaptic glycine and GABA receptors, respectively, and tetrodotoxin (TTX) to block axonal voltage-gated sodium channels, thereby preventing action potential propagation. Control conditions used D-2-amino-5- phosphonovaleric acid (APV) to inhibit glutamate binding to NMDAR (Zivanovic and Susic, 1998). We first characterised the changes in FFAs levels in response to glycine-induced cLTP by liquid chromatography tandem mass spectrometry (LC–MS/MS) using the FFAST derivatisation pipeline (Narayana et al, 2015; Wallis et al, 2021) (Fig. 1B,C). Induction of cLTP dramatically increased sFFAs, particularly myristic, palmitic and stearic acids. This is in good agreement with our previous reports that neuronal stimulation and memory acquisition both promote a marked increase in sFFAs (Akefe et al, 2024; Narayana et al, 2015; Wallis et al, 2021). Pharmacological inhibition of DDHD2 using KLH-45 completely ablated the cLTP-induced sFFA response, demonstrating that the PLA1 activity of DDHD2 was required for the response. To test the hypothesis that sFFAs are CoA-conjugated prior to being used to lipidate synaptic proteins, we used LC–MS/MS-based techniques to examine the formation of CoA conjugates of the three sFFAs over time. We found that cLTP drove significant increases in myristoyl CoA, and to a lesser extent stearoyl CoA immediately after (10 min) glycine treatment which rapidly re-equilibrated towards baseline (Fig. 1D). Examination of the relative molar concentrations of sFFA CoA conjugates at rest (APV treated) showed that myristoyl CoA only represent 0.3% of

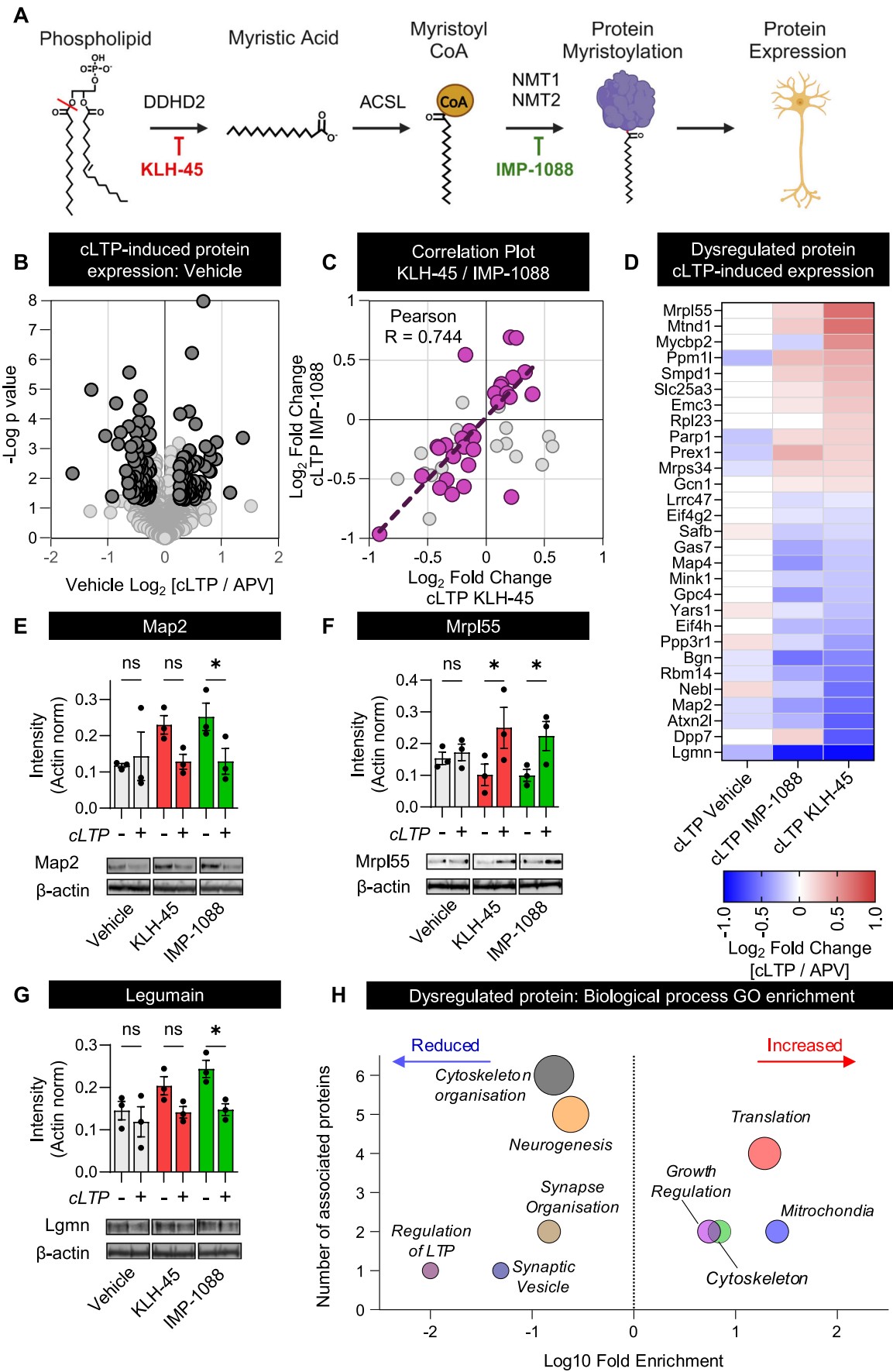

**Figure 2.  Treatment with the DDHD2 inhibitor KLH-45 and the NMT1/2 inhibitor IMP-1088 alters cLTP-induced protein expression, resulting in commonly dysregulated proteins.**

(A) Schematic displaying the proposed pathway through which DDHD2-derived myristic acid provides substrate for protein myristoyl modification. The effect of KLH-45 and IMP-1088 on this pathway is assessed using untargeted proteomics. (B) Volcano plot displaying cLTP-induced changes to protein expression compared to the APV-treated negative controls in Vehicle (DMSO)-treated cortical neurons. (C) Quadrant plot correlating proteins displaying a significant change in expression in response to KLH-45 and IMP-1088. Proteins with the same cLTP-induced expression as vehicle controls are coloured grey, while proteins dysregulated upon inhibitor treatment are coloured purple. Pearson correlation coefficient of dysregulated protein expression in response to IMP-1088 and KLH-45 treatment, $R = 0.744$. (D) Heatmap of the sensitivity to KLH-45 and IMP-1088 to cLTP-induced changes in protein expression. (E–G) Bar plots of western blot densitometry of a selection of such dysregulated proteins: Map2, Mrpl55 and Legumain (Lgmn) respectively. (H) Bubble plot displaying significantly enriched biological process GO terms in the inhibitor dysregulated proteins. The $\log_{10}$ fold enrichment of each term in the dysregulated proteins compared to the *Mus musculus* reference proteome is displayed on the *x*-axis. The number of dysregulated proteins associated with each term is indicated by bubble size and *y*-axis position. Data information: Experiments were performed using three biological replicates. (B) Volcano plot of protein fold change. The acceptance thresholds for significant change, dark grey circles, combined a $\log_2$ [cLTP/APV] fold change <0.3 with a *t*-test *p* value < 0.05. (C) Correlation plot of the cLTP-induced changes in protein expression that are sensitive to KLH-45 and IMP-1088 inhibitors. (D) Dysregulated protein expression (assessed by subtracting the cLTP-induced change in expression (Log$_2$ [cLTP/APV]) observed in inhibitor-treated neurons from the cLTP responses in vehicle controls). Proteins displaying values >3 or <0.33 were considered dysregulated in response to inhibitor treatment. (E–G) The error bar indicates SEM. Statistical analysis was performed using two-way ANOVA comparing cLTP and inhibitor effects. Pairwise significance of group means was performed using Fisher's least significant difference test. Map2 (E) two-way ANOVA: cLTP effect $p = 0.048$, Inhibitor (KLH-45 or IMP-1088) effect $p = 0.264$; Pairwise cLTP effect: Vehicle $p = 0.631$; KLH-45 $p = 0.075$; IMP-1088 $p = 0.036$. Mrpl55 (F) two-way ANOVA: cLTP effect $p = 0.009$, Inhibitor effect $p = 0.924$; Pairwise cLTP effect: Vehicle $p = 0.741$; KLH-45 $p = 0.018$; IMP-1088 $p = 0.041$. Lgmn (G) two-way ANOVA: cLTP effect $p = 0.005$, Inhibitor effect $p = 0.043$; Pairwise cLTP effect: Vehicle $p = 0.423$; KLH-45 $p = 0.071$; IMP-1088 $p = 0.01$. (H) GO biological process term enrichment was performed using Fisher's exact test comparing our dataset to the *Mus musculus* reference proteome from the PANTHER database. *$p < 0.05$ were considered significant.

the trio (myristoyl palmitoyl and stearoyl CoAs) while palmitoyl and stearoyl CoAs represent 52.2 and 47.5%, respectively (Fig. 1E). This ratio is changed upon cLTP induction especially for myristoyl CoA now representing 3.2%, whereas palmitic CoA and stearic CoA represent 40.1 and 56.7% respectively (Fig. 1E). This suggests that acyl CoA synthase (ACS) substrate specificity favours the generation of myristoyl CoA during cLTP. Measurement at 30 and 120 min post-cLTP showed a rapid re-equilibration of myristoyl CoA concentrations back towards baseline control conditions, suggesting that the myristoyl CoA generated in response to cLTP is rapidly metabolically utilised. The experiment was repeated in the presence or absence of the selective DDHD2 inhibitor KLH-45 (Inloes et al, 2014) which greatly reduced the sFFA CoA conjugates response to cLTP at the 10 min time point (Fig. 1F). Importantly, all responses were significantly reduced by DDHD2 inhibition (Fig. 1F). Chemical-LTP-induced myristoyl CoA production was greatly diminished in response to KLH-45 treatment but not completely ablated. This suggests that myristic acid derived from DDHD2 activity is the primary substrate for cLTP-induced myristoyl CoA production, although myristic acid from other sources might also contribute to the response.

Overall, our results demonstrate that sFFAs are generated by DDHD2 in response to cLTP, leading to the generation of CoA conjugates that are rapidly metabolically utilised. This suggests that activity-dependent generation of myristoyl CoA could underpin the plastic response.

## DDHD2 and NMT1/2 control the expression of a common set of synaptic proteins during cLTP

We next sought to determine whether DDHD2 and NMT1/2 play a larger role in shaping the plasticity of potentiated neurons (Fig. 2A). We hypothesised that if stimulation-induced myristic acid produced by DDHD2 is an essential precursor required to facilitate activity-induced protein myristoylation, then disruption of either DDHD2 or NMT1/2 activity should result in a common set of dysregulated proteins. To assess this, we performed proteomic profiling using high-resolution mass spectrometry (HRMS)

analysis. In order to allow sufficient time for responsive protein translation to occur, we carried out this analysis 120 min after cLTP with or without inhibitors. Cortical neurons were pretreated with either KLH-45 or IMP-1088, respectively, prior to cLTP. cLTP promotes the upregulation of 109 proteins and downregulation of 108 out of a total of 2596 proteins detected (Fig. 2B). Upon preincubation with IMP-1088, we detected 101 upregulated proteins and 113 downregulated proteins in response to cLTP out of 2596 total proteins. Upon preincubation with KLH-45, we detected 282 upregulated proteins and 237 downregulated proteins (out of 3686 proteins) in response to cLTP. The full list of up- and down-regulated proteins is included in Dataset EV1. De novo protein synthesis is intimately involved in the LTP plastic response (Cajigas et al, 2010), with alteration to protein expression leading to perturbed plasticity (Stanton and Sarvey, 1984). As DDHD2 provides myristic acid that is subsequently CoA conjugated, NMT1/2 and DDHD2 should act in the same pathway to promote protein myristoylation involved in downstream plastic responses. Inhibiting DDHD2 or NMT1/2 should therefore result in a similar pattern of proteomic disruption (Fig. 2A). To test this, we compared the change in protein expression induced by cLTP in neurons with those pretreated with either KLH-45 and IMP-1088. This revealed 29 proteins which highly correlated between the two inhibitor treatments (Fig. 2C). A heatmap comparing the cLTP-induced expression of these proteins in vehicle, KLH-45 and IMP-1088-treated neurons is displayed in Fig. 2D. We then selected Map2 (Fig. 2E), Mrpl55 (Fig. 2F) and Legumain (Lgmn) (Fig. 2G) and confirmed these expression changes by western blotting. The statistical significance of these results was determined using two-way ANOVA comparing cLTP and inhibitor treatment effects, showing significant change (Map2: cLTP effect $p = 0.048$, inhibitor effect $p = $ n.s.; Mrpl55 2 cLTP effect $p = 0.009$, Inhibitor effect $p = $ n.s.; Legumain cLTP effect $p = 0.005$, Inhibitor effect $p = 0.043$). Pairwise comparison of cLTP and APV mean densitometry intensity showed no change in the vehicle controls, while IMP-1088 treatment elicited a significant effect in all three proteins. This analysis yielded a significant effect in KLH-45-treated neurons for Mrpl55, while Legumain and Map2 returned *p* values of 0.071 and

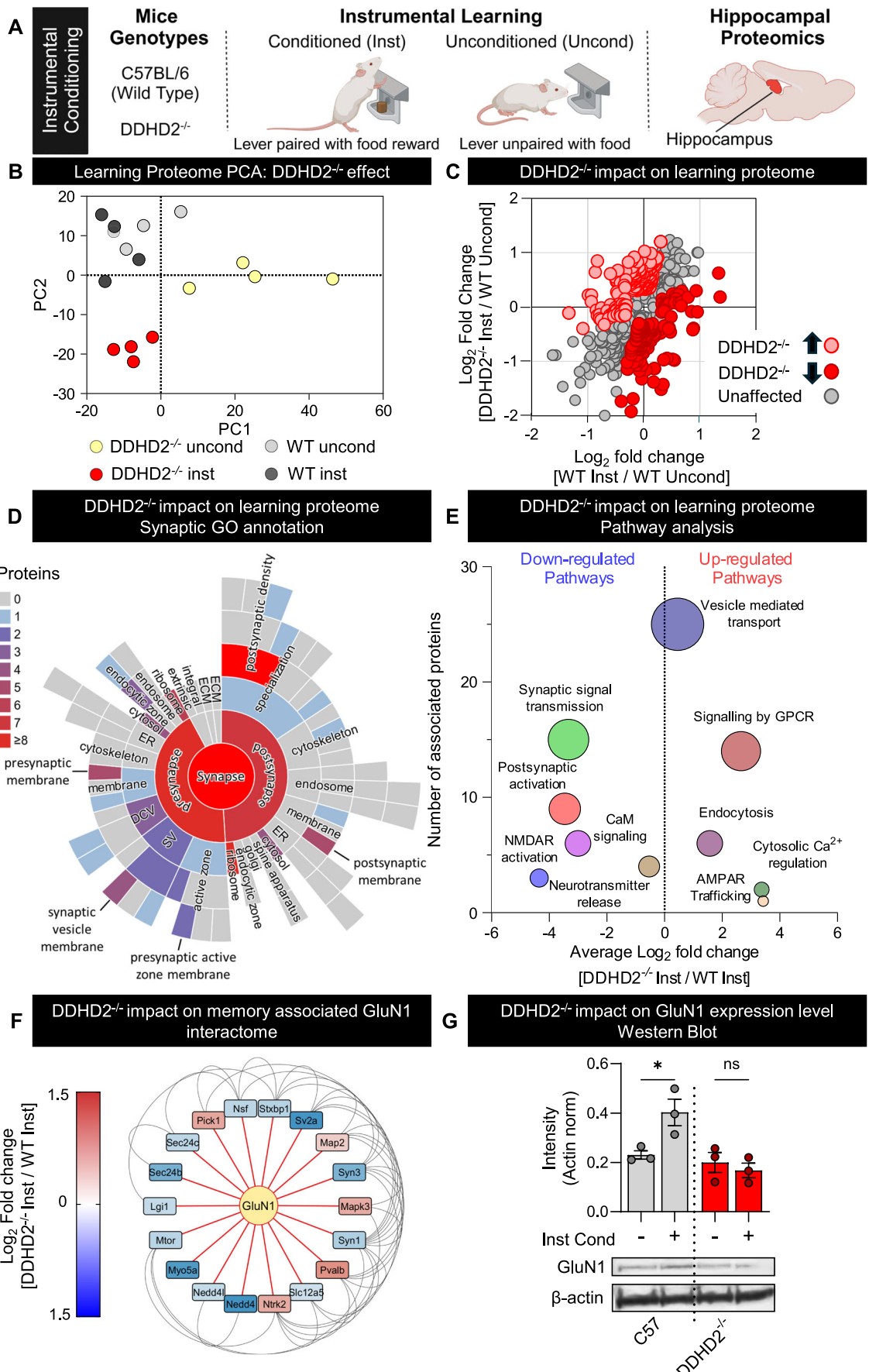

**A** Instrumental Conditioning

**Mice Genotypes**
C57BL/6 (Wild Type)
DDHD2⁻/⁻

**Instrumental Learning**
Conditioned (Inst) — Lever paired with food reward
Unconditioned (Uncond) — Lever unpaired with food

**Hippocampal Proteomics**
Hippocampus

**B** Learning Proteome PCA: DDHD2⁻/⁻ effect

- DDHD2⁻/⁻ uncond
- DDHD2⁻/⁻ inst
- WT uncond
- WT inst

**C** DDHD2⁻/⁻ impact on learning proteome

DDHD2⁻/⁻ ↑
DDHD2⁻/⁻ ↓
Unaffected

**D** DDHD2⁻/⁻ impact on learning proteome Synaptic GO annotation

**E** DDHD2⁻/⁻ impact on learning proteome Pathway analysis

**F** DDHD2⁻/⁻ impact on memory associated GluN1 interactome

**G** DDHD2⁻/⁻ impact on GluN1 expression level Western Blot

**Figure 3. DDHD2 knockout dysregulates learning-induced expression of synaptic proteins in the dorsal hippocampus.**

(A) Overview of instrumental conditioning of DDHD2$^{-/-}$ and C57BL/6 (Wild Type) mice. (B) PCA plot of hippocampal proteins displaying significantly altered expression in response to instrumental conditioning in either DDHD2$^{-/-}$ or wild-type mice. (C) Quadrant plot comparing the significant learning responsive hippocampal protein expression in DDHD2$^{-/-}$ and wild-type mice. Proteins coloured light red display increased expression upon ablation of DDHD2, while dark red coloured proteins have decreased expression. (D) Segmented sunburst plot displaying the synaptic GO cellular component terms of the above-mentioned responsive proteins (learning-induced proteins dysregulated by DDHD2 genetic ablation). (E) Bubble plot of significantly up- and down-regulated pathways when comparing instrumentally conditioned DDHD2$^{-/-}$ and wild-type hippocampal proteomes using reactome geneset analysis (RGSA). The average log$_2$ fold change of associated proteins is displayed on the x- axis. The number of proteins detected in the pathway is depicted by bubble size and position on the y-xis. (F) Network plot displaying the change in GluN1 NMDA receptor subunit interactome between DDHD2$^{-/-}$ and wild-type instrumentally conditioned mice. Box colour indicates Log$_2$ fold change expression, with red denoting increased expression and blue decreased expression. (G) Western blotting analysis of hippocampal lysates probed for GluN1. Bar plot of GluN1 band densitometry normalised to β-actin. Data information: Experiments were performed using four biological replicates. (B) The criteria for identifying proteins with significantly altered expression and PCA conditions were as described for Fig. 2. (C) Protein expression dysregulation was determined by subtracting the instrumental responsive change in DDHD2$^{-/-}$ animals by the expression change in wild-type controls. Proteins displaying values >3 or <0.33 were considered dysregulated in the hippocampus of DDHD2$^{-/-}$ animals. (E) Pathway enrichment analysis was performed using the RGSA PADOG algorithm. Pathways with $p$ values < 0.05 were considered significantly enriched. (F) Hippocampal proteins were considered differentially expressed between the DDHD2$^{-/-}$ and wild-type instrumentally conditioned animals if the $p$ value for the $t$-test comparison intensity values was ≤0.05 and log$_2$ fold change was ≥0.25. GluN1 interactors among the differentially expressed proteins was determined using the STRING protein interaction database with a combined interaction score cut off ≥0.4. (G) Western blotting experiments for the GluN1 NMDA receptor subunit were performed on hippocampal lysates obtained from three separate animals per treatment condition. Error bars indicate the SEM. Statistical analysis was performed using two-way ANOVA comparing (i) instrumental conditioning ($p = 0.101$), (ii) DDHD2 status ($p = 0.007$) and (iii) interaction effect ($p = 0.026$). Pairwise comparison instrumental condition effect group means was performed using Fisher's least significant difference test: C57BL/6 wild type $p = 0.012$; DDHD2$^{-/-}$ $p = 0.561$.

0.075, respectively, indicating a similar trend in the latter two proteins. The functional roles of these proteins were established using gene ontology term enrichment analysis (Thomas et al, 2022,) suggesting that blocking DDHD2 and NMT1/2 activity prevented cLTP-induced changes in synaptic vesicle cycle, regulation of synaptic plasticity, and synaptic organisation (Fig. 2H).

## DDHD2$^{-/-}$ alters hippocampal learning-induced protein expression, including NMDA receptor GluN1 subunit

We next characterised the impact of DDHD2$^{-/-}$ on learning-induced hippocampal protein expression in a mouse model. DDHD2$^{-/-}$ mice have previously been demonstrated to display profound learning and neuromuscular deficits and ablated sFFAs responses to learning (Akefe et al, 2024). DDHD2$^{-/-}$ and C57BL/6 wild-type (WT) control animals underwent a 14-day reward-based instrumental conditioning training regime, after which animals were euthanised and untargeted proteomic characterisation was performed on the dissected dorsal hippocampus (Fig. 3A). Learning-induced changes were broadly assessed through PCA comparison of the hippocampal protein expression profiles of the unconditioned (uncond) and instrumental conditioned (inst) animals (Fig. 3B), which clearly showed that the protein profiles of DDHD2$^{-/-}$ animals clustered away from those of the WT animals. Specific proteins undergoing expressional change between the experimental treatment conditions were determined using one-way ANOVA. While a significant number of proteomic changes seemed to be associated with DDHD2$^{-/-}$ status, instrumental conditioning drove changes to the WT hippocampal proteome, including the increased expression of the NMDA subunit GluN1, that were not evident in conditioned DDHD2$^{-/-}$ animals (Fig. EV1). A full list of differentially expressed proteins is presented in Dataset EV2. We performed a differential correlation analysis on the significantly altered proteins comparing learning-induced changes in the hippocampi of wild-type and DDHD2$^{-/-}$ animals. DDHD2$^{-/-}$ showed 186 learning-responsive proteins displaying increased expression and 171 displaying decreased expression (Fig. 3C). Gene ontology cellular component analysis (Fig. 3D) showed that 83

(22.6%) of these DDHD2$^{-/-}$ effected proteins are associated with the synapse, with 49 and 44 displaying pre- and post-synaptic association respectively. Notably, the specific region with the greatest concentration of DDHD2$^{-/-}$ affected proteins was the post-synaptic density with 20 proteins, followed by 12 synaptic vesicle proteins and 8 post-synaptic membrane proteins. This was supported by functional reactome geneset analysis (RGSA) showing that the DDHD2$^{-/-}$ altered proteins were involved in synaptic function and plasticity, including vesicular transport, synaptic transmission and NMDAR activation (Fig. 3E). To further explore this result, we performed a network analysis using the STRING database. We utilised first neighbour analysis to extract altered proteins from the DDHD2$^{-/-}$, which directly impact NMDAR GluN1 subunit function. Applying a combined interaction score greater then 0.4 revealed dysregulation of 19 interacting proteins (5 upregulated, 14 downregulated) (Fig. 3F). To confirm the impact of DDHD2$^{-/-}$, we performed western blotting analysis of the GluN1 NMDAR subunit which showed that knock-out of DDHD2 abrogated the learning responsive increase in GluN1 expression observed in the wild-type animals (Fig. 3G).

Finally, we investigated potential correlation between DDHD2 pharmacological inhibition (KLH-45) in vitro (Fig. 2) and genetic DDHD2$^{-/-}$ in vivo (Fig. 3). We found that both in vitro and in vivo inhibitions led to common activated or inhibited pathways with a number of proteins similarly affected including (but not limited to) Map2, Map4, Gas7 and RBM14 which are involved in (1) cytoskeleton organisation, (2) PI3K activated dendrite formation, and (3) co-factor in nuclear FFA receptor (PPAR) activation (DeGiosio et al, 2022; Firmin et al, 2017; Khanal et al, 2023; Nishida et al, 2023) (Fig. EV2).

## N-terminal myristoylation is unaffected by cLTP in vitro and instrumental conditioning in vivo

The N-terminal myristoyl modification is classically added co-translationally to glycine immediately after enzymatic removal of the initiator methionine on the nascent protein. To explore whether cLTP could promote N-terminal myristoylation, thereby mediating

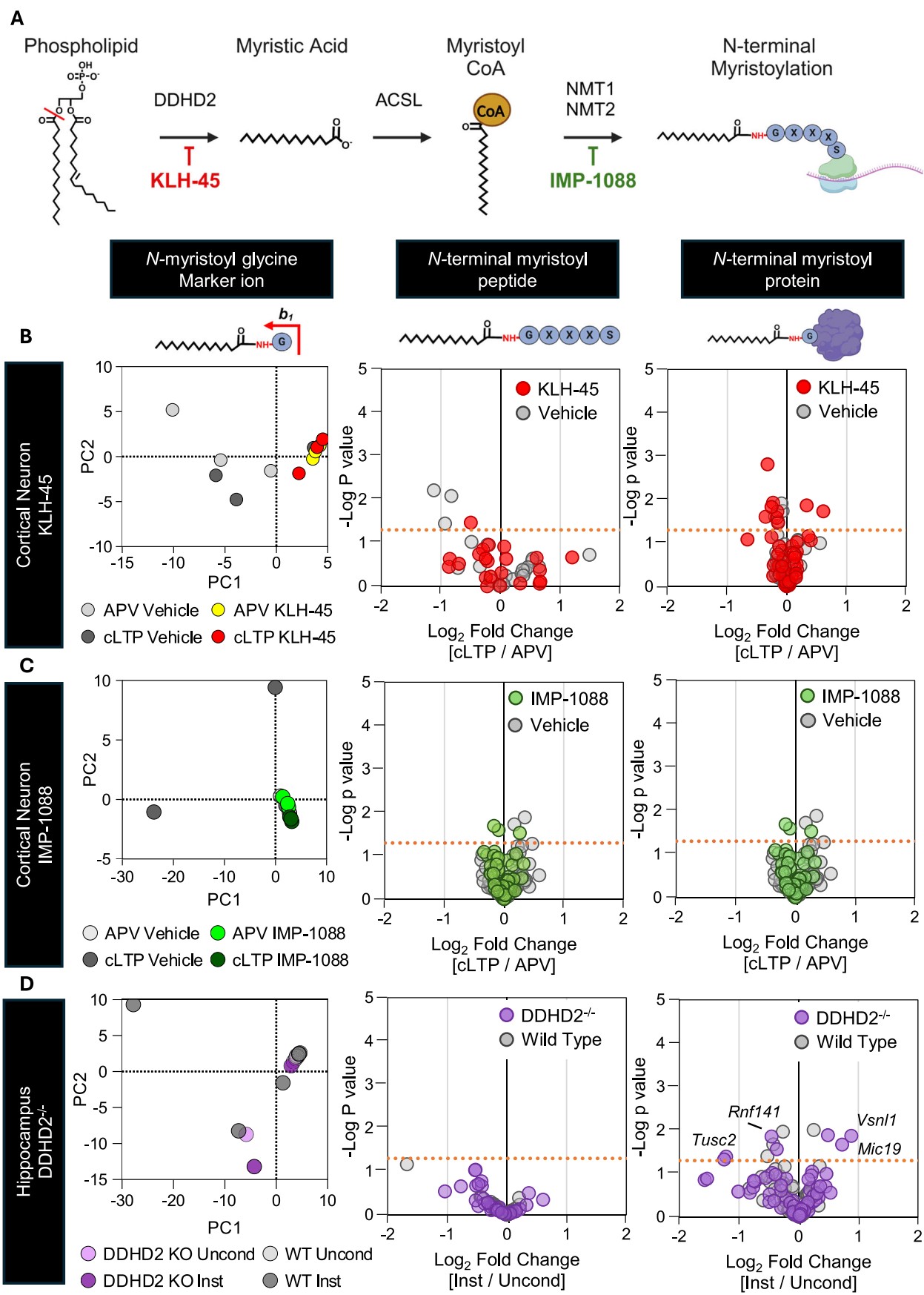

**Figure 4.  cLTP in vitro or DDHD2$^{-/-}$ in vivo does not instigate a significant change in the formation of new *N*-terminal myristoyl modifications.**

(A) Schematic displaying the proposed pathway through which DDHD2 and NMT1/2 work sequentially to drive *N*-terminal myristoyl modification. (B, C) Assessment of *N*-terminal myristoylation in cultured cortical neurons (DIV 21) 2 h post-cLTP induction in the continuing presence of either (B) KLH-45 (25 nM 4 h preincubation) or (C) IMP-1088 (500 nM 1 h preincubation). Protein extraction, digestion and LC–MS analysis were performed as described in Methods. Left panels – *N*-myristoyl glycine fragment marker ion analysis. PCA plot of precursor ion *m/z*, intensity and chromatographic retention time for MS/MS spectra containing the 268.23 *m/z* marker. Middle panels – Target *N*-terminal myristoyl peptide MRM analysis. Overlaid volcano plot displaying the relative log$_2$ cLTP-induced fold change in the targeted 27 *N*-terminal myristoylated proteins in vehicle and inhibitor-treated neurons. Right panels – Untargeted *N*-terminal myristoylated protein analysis. Overlaid volcano plot displaying the relative log$_2$ cLTP-induced fold change of proteins containing an *N*-terminal myristoyl modification in vehicle and inhibitor-treated neuronal cultures. (D) Changes in the hippocampal proteome of instrumental and unconditioned DDHD2$^{-/-}$ and wild-type mice. Data Information: All experiments were performed using 3 independent biological replicates. (B–D) PCA plot of precursor ion *m/z*, intensity and chromatographic retention time for MS/MS spectra containing the 268.23 *m/z* marker ion was extracted from HRMS datasets. For volcano plots, statistical significance was assessed using a *t*-test, with $p < 0.05$ considered significant. The significant -log$_{10}$ *p* value (1.3) is indicated by an orange dotted line on all volcano plots.

the proteomic phenotypes observed upon NMT1/2 inhibition, we used three different LC–MS-based approaches to assess cLTP-responsive changes of this modification: (1) Expression changes in *N*-terminally myristoylated proteins (as annotated in UniProt (Consortium, 2021)) were assessed using untargeted HRMS analysis to measure levels of non-myristoylated proteotypic peptides; (2) A novel tandem MS assay for detection of a unique myristoylglycine fragment ion to allow database-independent qualitative assessment of the *N*-terminal myristoylation state of the proteome. Myristoyl glycine marker ion data were clustered using principal component analysis (PCA) to profile the *N*-terminal myristoylome in each treatment condition; (3) A multiple reaction monitoring (MRM) LC–MS assay, which specifically quantified the *N*-terminally myristoylated tryptic peptide from 27 neuronal proteins (Appendix Table S1). We used the above mass spectrometry approaches 120 min after glycine stimulation in cLTP-induced neurons, to allow sufficient time for synthesis of new protein with new co-translational myristoylation events. KLH-45 and IMP-1088 inhibition were used to determine the degree to which DDHD2-derived myristoyl CoA is required for cLTP-induced *N*-terminal myristoylation. Comparison of the effects of inhibition of DDHD2 and NMT1/2 was used to provide insight into the relative contribution of these enzymes to the process (Fig. 4A). PCA was performed on *N*-terminal myristoyl glycine fragment marker ion data extracted from untargeted proteomics analysis. No clustering could be detected in response to either cLTP stimulation or inhibitor treatment (KLH-45: Fig. 4B left; IMP-1088: Fig. 4C left) and vehicle controls. Similarly, targeted quantitation of *N*-terminally myristoylated peptides showed little to no change within the acceptance threshold (Log$_2$ 0.25-fold change) of cLTP-induced in the presence of either KLH-45 (Fig. 4B centre) or IMP-1088 (Fig. 4C Centre) versus treatment controls. Examination of proteins annotated as containing an *N*-terminal myristoyl modification showed no cLTP-induced change within acceptance criteria in the presence or absence of either KLH-45 (Fig. 4B right) or IMP-1088 (Fig. 4C right). As inhibition of NMT1/2 activity using IMP-1088 should effectively stop any new myristoylation events, this result suggests that *N*-terminal myristoylation is stable over the 2 h testing period and did not undergo significant cLTP-induced change.

To examine if stable *N*-terminal myristoylation occurred in vivo, we applied the three *N*-terminal myristoylation testing paradigms to dorsal hippocampal protein extracts of instrumentally conditioned and control cohorts of 12 mo DDHD2$^{-/-}$ and WT mice which we had previously sampled for proteomics (Fig. 3) and lipidomic analysis

(Akefe et al, 2024). Similar to the result observed upon cLTP induction in vitro, no PCA clustering of the myristoyl glycine marker (Fig. 4D left) or change in *N*-terminal myristoyl modified domain (Fig. 4D centre) were observed in the learners versus non-learners hippocampi. Seven myristoylated proteins displayed differential learning responsive regulation in DDHD2$^{-/-}$ animals out of the 67 detected in the dataset (Fig. 4D right). However, these proteins were unchanged with instrumental conditioning in wild-type animals. Given the broad alterations to the hippocampal proteome observed in response to genetic ablation of DDHD2 (Fig. EV1), the minor changes observed here is unlikely to be due to *N*-terminal myristoylation status.

Taken together, these results indicate that *N*-terminally myristoylated proteins are stable and largely unaffected by either cLTP in vitro or operant conditioning in vivo. Furthermore, *N*-terminal myristoylation was unaffected by either acute pharmacological inhibition or chronic genetic knock-out of DDHD2, suggesting that myristic acid and myristoylCoA generated during synaptic plasticity and memory is not required for *N*-terminal myristoylation.

## Lysine myristoyl transferase activity alters the membrane localisation of neuronal effector proteins in an activity-dependent manner

Having shown that *N*-terminal myristoylation is largely unaffected by synaptic plasticity, we sought to determine if lysine myristoylation could instead undergo change upon neuronal activation. Due to the technical limitations of directly detecting lysine myristoylated peptides, we instead used an indirect approach. LTP has previously been reported to induce membrane translocation of synaptic effector proteins over short time scales, enabling initiation of plasticity (Angenstein et al, 1994; Mori et al, 2014). To explore the role of lysine myristoylation in this process, we performed subcellular fractionation and generated low- and high-density membrane fractions via ultracentrifugation. These were carried out immediately following cLTP to minimise the effects of protein synthesis or degradation in the presence or absence of IMP-1088 and KLH-45 (Fig. 5A,B). We used subcellular markers to confirm that the high-density fraction contained the plasma membrane, Golgi and mitochondrial membranes, whereas the low-density fraction was enriched in endosomes and synaptic vesicles (Fig. 5C). The scarcity of cortical neuron material obtained did not allow for further separation of our fractions. Untargeted proteomics analysis was performed on the high-density fraction (enriched in plasma membrane proteins) to assess potential recruitment/enrichment and sensitivity to inhibitor treatment. By identifying the membrane proteome that is sensitive to

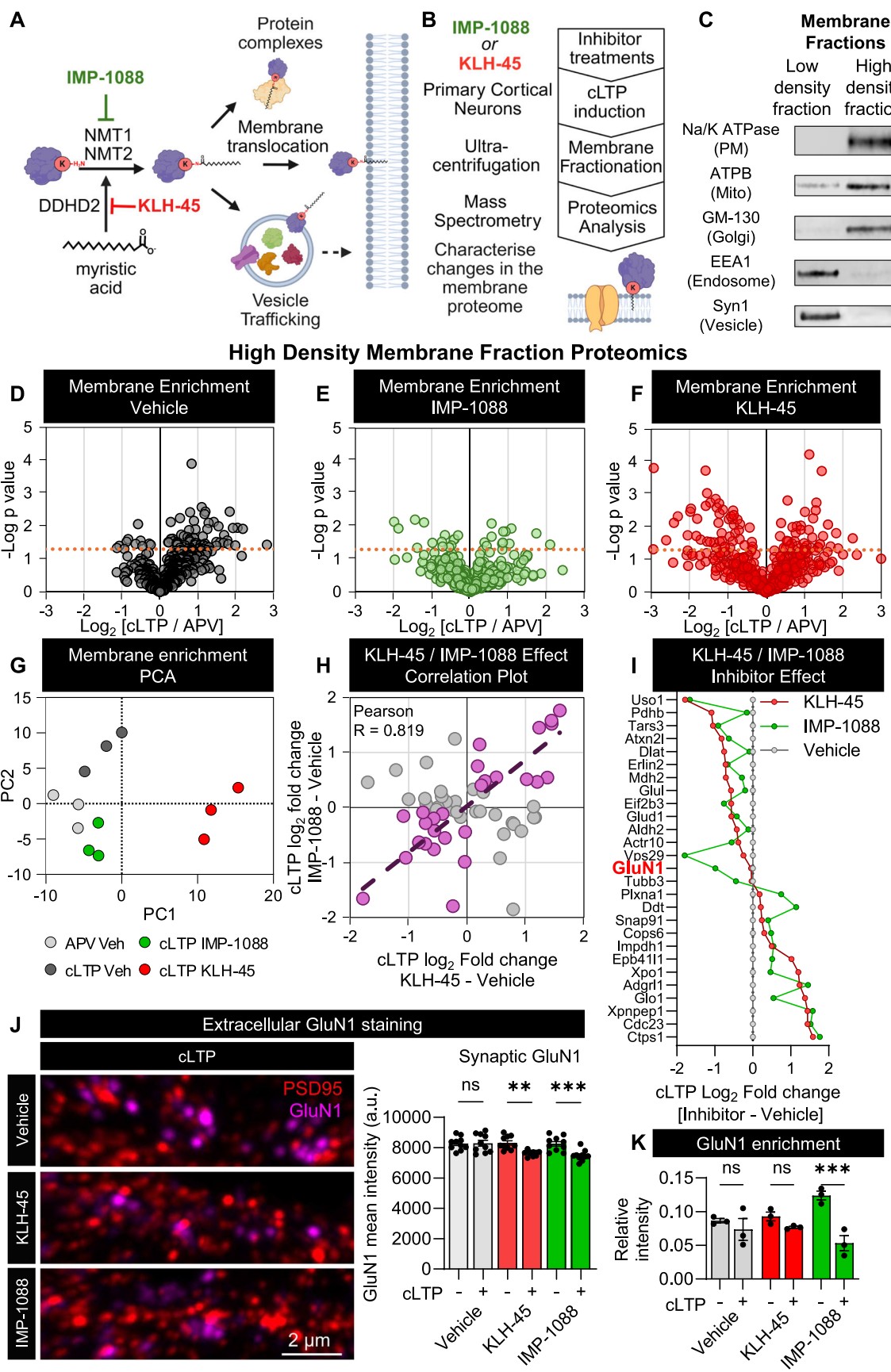

**High Density Membrane Fraction Proteomics**

**Figure 5. Both NMT1/2 and DDHD2 are essential for proper protein membrane partitioning during LTP.**

(A) Schematic displaying the proposed pathway impacting protein enrichment in membranes based on pharmacological inhibition of DDHD2 (KLH-45) and NMT1/2 (IMP-1088). (B) Proteomics analysis of isolated cortical neuron membrane fractions immediately after cLTP in the presence of NMT1/2 inhibitor IMP-1088 and the DDHD2 inhibitor KLH-45. (C) Western blot analysis for subcellular markers in the low- and high-density membrane fractions generated using ultracentrifugation. (D–F) Volcano plots showing cLTP-induced protein enrichment and depletion compared to APV-treated controls in vehicle (dark grey), IMP-1088 (green) and KLH-45 (red) treated cortical neurons, respectively. Positive values indicate enrichment in the high-density membrane fraction whereas negative value indicate a depletion. The $y$-axis indicates the $-\log_{10}$ Student $t$-test $p$ values, with values exceeding 1.3 considered significant. (G) Principal component analysis plot of proteins displaying altered membrane localisation as determined by one-way ANOVA. (H) Quadrant plot comparing the effect of IMP-1088 and KLH-45 treatment on proteins displaying disrupted enrichment in high-density membrane fractions compared to the controls (cLTP $\log_2$ fold change observed in the inhibitor – cLTP $\log_2$ fold change observed in vehicle). Correlation analysis identified proteins with similar inhibitor-induced changes upon IMP-1088 and KLH-45 treatment (purple data points). Proteins with poor correlations between inhibitor effects are represented in grey. (I) Line plot displaying the relative protein expression changes of highly correlated inhibitor-effected proteins in cLTP-induced enrichment in high-density membrane fractions compared to the vehicle treated neurons. Negative values depict a decreased membrane enrichment compared to the cLTP effect in vehicle controls, whereas positive values indicate an increased enrichment. (J) GluN1 synaptic surface expression analysis using a GluN1 extracellular Ab (magenta) and a PSD95 intracellular Ab (red) in neurons 10 min post-cLTP or APV treatment. Bar plot: KLH-45 (red) and IMP-1088 (green) treatments reduce GluN1 surface expression following cLTP. (K) Bar plot displaying the relative intensity of the NMDA receptor GluN1 subunit detected in the high-density membrane fraction from each treatment condition using untargeted proteomics. Data Information: Experiments were performed using three biological replicates. (D–F) Statistical analysis of cLTP-induced changes in high-density membrane fraction enrichment was performed using pairwise analysis of cLTP/APV signal intensities for each treatment condition (vehicle, IMP-1088 and KLH-45 treated) by Student $t$-test. $P$ values $\leq 0.05$ ($\geq 1.3$ $-\log_{10}$) and $\log_2$ fold change $\geq 0.25$ were considered significant. (G) Principal component analysis was performed on proteins displaying differential membrane partitioning as determined by one-way ANOVA, where $p$ values of $\leq 0.05$ were considered significant. PCA compared the highest 2 Eigenvalues >1 as the two plotted dimensions. (H) One-way ANOVA significant proteins ($p \leq 0.05$) displaying correlated inhibitor-induced disruption to enrichment in high-density membrane fractions was determined as follows: The effect of each inhibitor was determined by subtracting the $\log_2$ relative change in protein intensity in cLTP vehicle from relative cLTP-induced change observed in the inhibitor-treated neurons ($\log_2$ [cLTP veh/APV veh] - $\log_2$ [cLTP inhibitor/APV veh]). Proteins displaying an inhibitor effect $\geq 0.25$ were considered to have altered enrichment in the high-density membrane fraction. Commonalities between inhibitor treatments were determined by dividing the IMP-1088 effect by the KLH-45 effect and selecting proteins displaying a ratio between 0.3 and 3. Correlations were statistically confirmed using the Pearson correlation test, with R coefficients greater than 0.7 considered highly correlated. (J) Analysis was performed over two replicates of cLTP treatments with ten regions of interest (ROIs) analysed per condition. Error bars indicate the SEM. Statistical significance was determined using an ordinary one-way ANOVA with post hoc pairwise analysis using Šídák's multiple comparisons test between APV and cLTP for each group. cLTP-induced change $p$ values: Vehicle $p = 0.998$; KLH-45 treated $p = 0.002$; IMP-1088 treated $p = 0.0006$. Asterisks indicate multiple comparisons between APV and cLTP treatment, **$p < 0.01$, ***$p < 0.001$. (K) Error bars indicate the SEM. Statistical analysis was performed using two-way ANOVA comparing (i) cLTP effect ($p = 0.0007$), (ii) inhibitor treatment ($p = 0.6543$) and (iii) interaction effect ($p = 0.012$). Pairwise comparison of the cLTP of each treatment group was performed using Fisher's least significant difference test: Vehicle $p = 0.331$; KLH-45 $p = 0.229$; IMP-1088 $p = 0.0001$. ***$p \leq 0.0001$.

DDHD2 and NMT1/2 inhibitors immediately after cLTP stimulation (10 min), we hoped to capture changes that are mainly dependent on post-translational lysine myristoylation. We indeed found that cLTP triggered a clear increase in the number of proteins detected in the high-density membrane fraction (Fig. 5D). This increase was sensitive to NMT1/2 inhibition suggesting that de novo lysine myristoylation is a requirement for cLTP-induced protein levels (Fig. 5E). Similarly, DDHD2 inhibition altered the membrane recruitment response to cLTP suggesting that myristic acid generated during cLTP was being used to drive de novo myristoylation (Fig. 5F). The full list of proteins displaying differential membrane enrichment in response to cLTP and inhibitor treatment is displayed in Dataset EV3. The impact of inhibitor treatment on protein levels in the high-density membrane fraction was determined using PCA (Fig. 5G). This showed that IMP-1088 and KLH-45 treatment resulted in a marked difference between both cLTP and APV vehicle controls, though the KLH-45 treatment resulted in the more pronounced effect, indicating the involvements of DDHD2 in localising proteins to these membranes extended beyond provision of substrate for protein myristoylation. To explore the specific proteins driving this effect, we performed ANOVA analysis to determine which proteins were differentially expressed in the high-density membrane fraction between the vehicle, KLH-45 and IMP-1088 treatments. We then performed a correlation analysis of inhibitor-affected proteins between the KLH-45 and IMP-1088 treatments (Fig. 5H). Of the 58 proteins with significant change compared to vehicle controls, we found a significant correlation between the effect of KLH-45 and IMP-1088 on a number of proteins, further indicating a common pathway. Examination of the proteins with similarly disrupted membrane enrichment upon inhibition of DDHD2 and NMT1/2 activity revealed key proteins necessary for

synaptic plasticity including the NMDA receptor subunit GluN1 and VPS29 (Fig. 5I). To determine whether the synaptic surface expression of NMDA receptor GluN1 subunit was sensitive to inhibitor treatment upon cLTP induction, we performed immunofluorescent staining for both the extracellular domain of GluN1 and the intracellular domain of PSD95. In support of our proteomics findings showing significantly decreased GluN1 (Fig. 5K) in the high-density membrane fraction, we found that the synaptic surface expression of GluN1 is reduced in cLTP-induced neurons treated with DDHD2 and NMT1/2 inhibitors (Fig. 5J). Altogether, our data indicate that DDHD2 and NMT1/2 work in concert to drive post-translational myristoylation, leading to altered membrane association of key proteins involved in synaptic function and plasticity.

## Myristoylation is necessary for cLTP-induced dendritic spine formation

One of the morphological correlates of LTP is the generation of new dendritic spines and their maturation (Borczyk et al, 2019). We therefore tested the effect of cLTP on spinogenesis and morphology and asked whether KLH-45 and IMP-1088 treatment could prevent cLTP-induced structural changes. Mature hippocampal neurons (21 days in vitro, DIV 21) were pretreated with either KLH-45 or IMP-1088 or vehicle prior to induction of cLTP or APV control. We found a significant increase in the average number of spines (Fig. 6A–C), their areas (Fig. 6B,D,G) and width (Fig. 6B,E,H), 2 h post-cLTP. These effects were completely blocked by both KLH-45 and IMP-1088 (Fig. 6B–H). Spine length remained unaffected by these treatments (Fig. 6F).

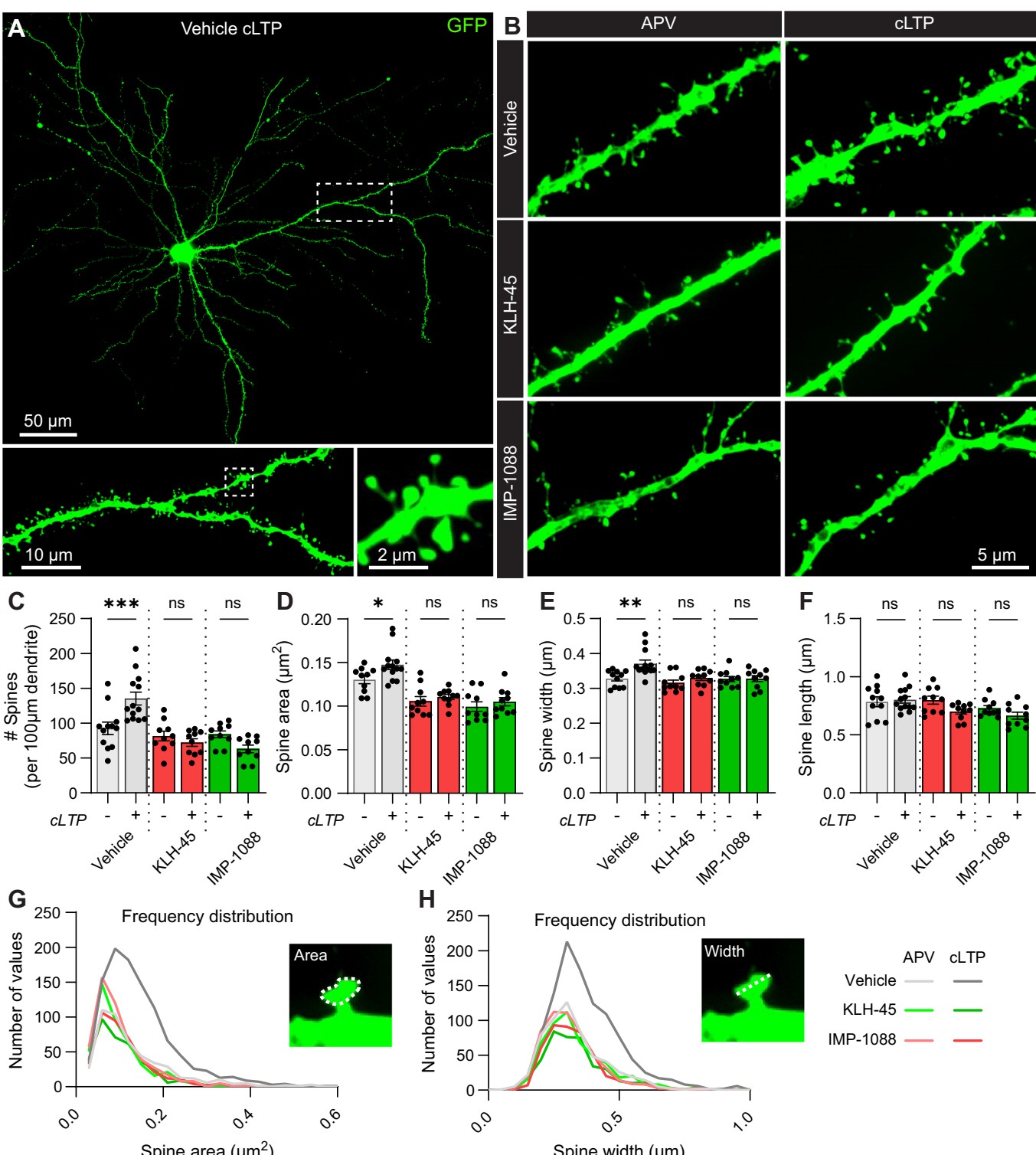

**Myristoylation is essential for field-induced potentiation in mouse hippocampal brain slices**

To determine how DDHD2 supports critical neuronal mechanisms underlying learning, plasticity and memory, we assessed whether DDHD2 and NMT1/2 deficiency affect theta-burst stimulation

(TBS)-induced LTP (Rodrigues et al, 2021). We performed local field potential recordings in hippocampal mouse brain slices (CA1, stratum radiatum, Fig. 7A) in artificial cerebrospinal fluid (ACSF, control), and in slices pretreated with KLH-45. Potentiation induced in untreated control slices was observed as an increase in the slope of the field excitatory post-synaptic potential (fEPSP)

**Figure 6.   DDHD2 and myristylation regulate cLTP-induced spinogenesis and spine maturation.**

Cultured EGFP-transfected (green) hippocampal neurons (DIV 21) were pretreated with either vehicle, DDHD2 inhibitor (KLH-45, 25 nM for 4 h) or NMT1/2 inhibitor (IMP-1088, 500 nM for 1 h) and subjected to cLTP treatment or APV control (for 10 min). Cells were briefly rinsed, incubated in neurobasal media for 2 h to allow cLTP-induced spine formation and subsequently fixed for imaging. (A) Example of maximum projection 3D Z-stack image of a mature primary hippocampal neuron 2 h after cLTP induction, with two zoomed-in panels showing dendritic spines. (B) Example maximum projection 3D Z-stack images of neurons pretreated with either vehicle, KLH-45 or IMP-1088. (C) Bar plot of the number of dendritic spines per 100 μm of dendrite length showing an increase in spines in response to cLTP in the vehicle control group, which is ablated in the KLH-45 and IMP-1088 treated groups. (D) Bar plot showing an increase in average dendritic spine head area in response to cLTP in vehicle controls, which is absent in inhibitor-treated groups. (E) Bar plot of average spine width, which is increased with cLTP but not with inhibitor treatment. (F) Bar plot of average spine length showing no change with cLTP or inhibitor treatment. (G) Frequency distribution plot of spine area showing an increase in large spines with cLTP in vehicle controls compared to inhibitor-treated groups. (H) Frequency distribution plot of spine width showing an increase in spine width with cLTP in vehicle controls compared to inhibitor-treated groups. Data information: Analysis was performed over three replicates of cLTP treatments with 3–4 cells analysed per replicate ($n \geq 10$). All error bars indicate the SEM. (C) Spine counts were performed across an entire dendrite (primary to tertiary) and normalised to spines per 100 μm length. Chemical-LTP induced change $p$ values: Vehicle $p = 0.0002$; KLH-45 treated $p = 0.784$; IMP-1088 treated $p = 0.179$. (D–H) Spine head area, width and length were traced on 3 ROIs per cell, dendrite averages of each ROI shown in bar plots, raw data of each spine plotted in frequency distributions. Statistical significance was determined using ordinary one-way ANOVA with post hoc pairwise analysis using Šídák's multiple comparisons test between APV and cLTP for each group. Spine number (C) $p$ values: Vehicle $p = 0.0002$; KLH-45 treated $p = 0.784$; IMP-1088 treated $p = 0.179$; Spine area (D) $p$ values: Vehicle $p = 0.0402$; KLH-45 treated $p = 0.870$; IMP-1088 treated $p = 0.820$. Spine width (E) $p$ values: Vehicle $p = 0.0023$; KLH-45 treated $p = 0.643$; IMP-1088 treated $p = 0.968$. Spine length (F) $p$ value: Vehicle $p = 0.970$; KLH-45 treated $p = 0.103$; IMP-1088 treated $p = 0.435$. *$p < 0.05$, **$p < 0.01$, ***$p < 0.001$.

following TBS (Fig. 7B–D). Conversely, we found that in the absence of DDHD2 activity LTP induction was prevented in KLH-45 treated slices, compared to control (Fig. 7). To confirm this was indeed due to the absence of DDHD2 activity rather than off-target inhibitor effects we then repeated the experiment in brain slices from genetically deficient DDHD2$^{-/-}$ animals (Fig. 7A) which also displayed a lack of potentiation post-TBS compared to WT slices. Finally, to examine the necessity of new protein myristoyl modification on this effect, we pretreated WT hippocampal slices with IMP-1088, resulting in no significant potentiation, compared to control (Fig. 7B–D). These results indicate that both DDHD2 and NMT1/2 are essential for LTP formation in the CA1 region of the hippocampus, and their blockade or ablation leads to an inability of the hippocampal circuitry to potentiate (Fig. 7E).

## Discussion

Myristic acid and other saturated FFAs are produced in key brain regions during fear (Wallis et al, 2021) and instrumental conditioning via the action of PLA1 DDHD2 (Akefe et al, 2024). We now provide evidence that cLTP also promotes myristic acid production by DDHD2 in mature hippocampal neurons, leading to the generation of myristoyl CoA, which is in turn used as substrate for NMT1/2-mediated de novo K-myristoylation (Fig. 7E). We show that the proteome changes induced by cLTP in vitro and by instrumental conditioning in vivo are sensitive to genetic and pharmacological inhibition of DDHD2 and NMT1/2. We found that the changes in the number of dendritic spines and their morphology induced by cLTP are also sensitive to DDHD2 and NMT1/2 inhibition. Finally, we demonstrate that DDHD2 and NMT1/2 control the establishment of LTP in brain slices. This adds to our recent report demonstrating that learning and memory is underpinned by DDHD2 which is targeted to the synapse via an interaction with Munc18-1/syntaxin binding protein 1 (STXBP1) (Akefe et al, 2024), a protein critically involved in SNARE-mediated neurotransmitter release (Chai et al, 2016; Han et al, 2010; Han et al, 2011; Han et al, 2009; Malintan et al, 2009).

## DDHD2-derived myristic acid produced in response to cLTP is converted to myristoyl CoA

As these experiments were performed in neuronally enriched cultures, where β oxidation likely only supplies a small proportion of the cells' energy demands, it is likely that the cLTP-generated myristoyl CoA was utilised for anabolic metabolism rather than supplying substrate for oxidative phosphorylation. Catabolic metabolism from supporting glia may contribute to some of the myristoyl CoA usage. Acyl CoAs are amphipathic molecules and excess accumulation can cause membrane damage due to detergent effects and intracellular micelle formation (Hsu and Powell, 1975), an effect that will be far more prominent in long-chain acyl CoAs due to their hydrophobicity. This detergent effect can alter membrane permeability (Banhegyi et al, 1996), and homoeostasis. This creates a cellular impetus for the cell to prevent the accumulation of acyl CoAs, which could explain the transient nature of the myristoyl CoA response to cLTP.

Surprisingly, the myristoyl CoA generated in response to cLTP showed a greater relative increase than palmitoyl and stearoyl CoA. We previously demonstrated a high degree of selectivity for myristoyl-containing phospholipid substrates during fear memory acquisition (Wallis and Meunier, 2024; Wallis et al, 2021) and instrumental conditioning (Akefe et al, 2024). Our current findings show that myristic acid is preferentially (220% increase) converted to myristoyl CoA in response to cLTP, an effect sensitive to DDHD2 inhibition, suggesting that acyl CoA synthetases (ACS) could also display specificity for myristic acid. As a comparison, the far more abundant palmitoyl and stearyl CoA only increase by 10 and 23%, respectively. Such selectivity by ACSs for myristic acid advocates for its role in synaptic plasticity and memory formation. In rodents, the ACSs encompass 25 different proteins with varying substrate affinities and tissue expression, responsible for shuttling acyl substrates to specific sites of metabolism (Grevengoed et al, 2014). Of these, the five long-chain acyl CoA synthetases (ACSL) (substrate specificity for C12–C20 FFAs) and the three bubblegum acyl CoA synthases (ACSBG) (substrate specificity for C14–C24 FFAs) have substrate specificities encompassing myristic acid (Soupene and Kuypers, 2008). All ACSL and ACSBG isoforms have been reported to be expressed in the brain, and ACSBG1 and

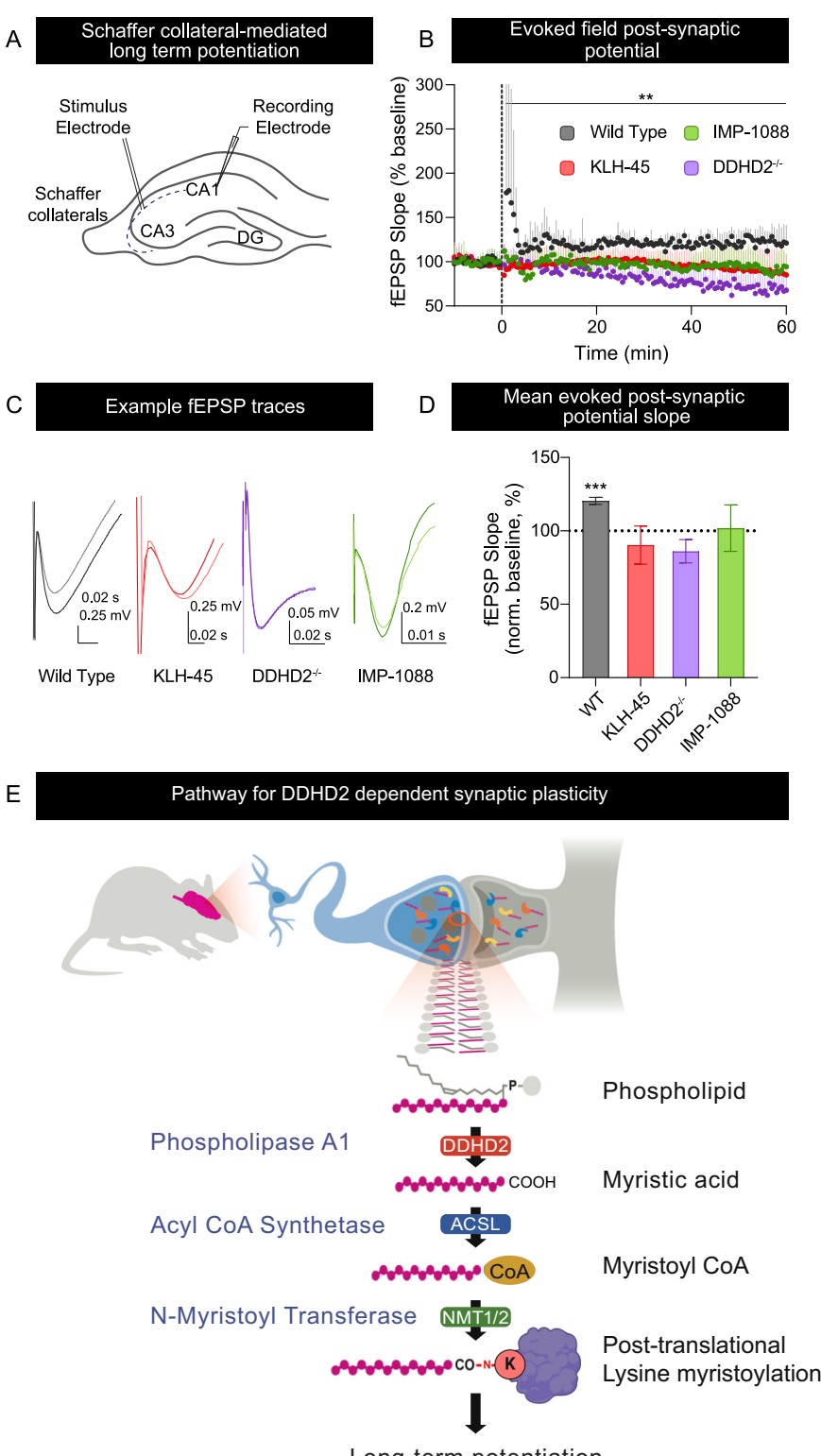

**A** Schaffer collateral-mediated long term potentiation

Stimulus Electrode

Recording Electrode

Schaffer collaterals

CA1

CA3

DG

**B** Evoked field post-synaptic potential

**

Wild Type

KLH-45

IMP-1088

DDHD2⁻ᐟ⁻

**C** Example fEPSP traces

Wild Type

KLH-45

DDHD2⁻ᐟ⁻

IMP-1088

**D** Mean evoked post-synaptic potential slope

***

WT

KLH-45

DDHD2⁻ᐟ⁻

IMP-1088

**E** Pathway for DDHD2 dependent synaptic plasticity

Phospholipid

Phospholipase A1 — DDHD2

Myristic acid

Acyl CoA Synthetase — ACSL

Myristoyl CoA

N-Myristoyl Transferase — NMT1/2

Post-translational Lysine myristoylation

Long-term potentiation

ACSL6 are specifically enriched in the brain compared to peripheral tissues (Fernandez and Ellis, 2020). Importantly, ACSL6⁻ᐟ⁻ mice strains display memory and motor function deficits (Fernandez et al, 2021) in good agreement with our study and altered dopaminergic signalling (Fernandez et al, 2023). We found that ACSL6 displayed an increase in expression in wild-type animals in response to instrumental conditioning, with no change evident in DDHD2⁻ᐟ⁻ animals. This advocates for a role of ACSL6 in processing DDHD2-derived myristic acid. Unconjugated myristic acid could also directly contribute to synaptic plasticity by

**Figure 7. Myristoylation is required for long-term potentiation in the rodent hippocampus.**

Electrophysiological extracellular recordings were taken from brain slices of wild-type C57BL/6 or DDHD2$^{-/-}$ mice. C57BL/6 brain slices were further treated with either KLH-45 (25 nM 4 h pretreatment), IMP-1088 (500 nM 1 hr pretreatment), or left as untreated controls (Wild Type, WT). (A) Schematic of Schaffer collateral-mediated long-term potentiation (LTP) recording in coronal hippocampal slices, with a recording electrode in the stratum radiatum of CA1, and a bipolar stimulating electrode at the Schaffer collaterals in CA2/3. (B) Measured slope of the evoked field post-synaptic potential (fEPSP) over time, prior to and following a theta-burst stimulation to evoke LTP (mixed effects model analysis): Represented as mean + SEM. (C) Example traces of fEPSP pre- (lighter) and post-theta-burst (darker), across all four conditions. (D) Mean evoked post-synaptic potential slope across 50 min following theta-burst stimulation, normalised to the baseline slope pre-stimulation. (E) Schematic of the proposed DDHD2 and NMT1/2 pathway for synaptic plasticity and memory. Data Information: Experiments were performed using the following number of replicates. Wild Type: $n = 4$ slices, 4 animals; KLH-45: $n = 3$ slices, 3 animals; DDHD2$^{-/-}$: $n = 4$ slices, 4 animals; IMP-1088: $n = 3$ slices, 3 animals). (B) Statistics were performed using the mixed effect model analysis F (conditions: 3, 10) = 7.068, $p = 0.0078$. **$p \le 0.01$. Data points represent the mean + SEM. (D) Statistical analysis performed using the Student's $t$-test comparing the mean evoked response slope to the baseline pre-stimulation slope for each condition (Wild Type $p = 0.000123$; KLH-45 $p = 0.498$; DDHD2$^{-/-}$ $p = 0.129$; IMP-1088 $p = 0.912$). ***$p \le 0.001$. Error bars indicate ±SEM.

acting as a lipid messenger. Both FFAR1 and FFAR4 cell surface receptors bind FFAs, leading to activation of protein kinase C (PKC) and phosphoinositide 3-kinase (PI3K)/AKT signalling pathways (Pyo et al, 2022), with knock out studies targeting these receptors showing memory and behavioural impairments in rodents (Falomir-Lockhart et al, 2019). Similarly, myristic acid could bind to PPAR nuclear transcription factors. PPARα has been shown to act as a mediator of hippocampal plasticity and memory (Roy and Pahan, 2015).

## Treatment with DDHD2 and NMT1/2 inhibitors disrupts cLTP-responsive protein expression

LTP has been associated with dramatic changes to the synaptic proteome (Diering and Huganir, 2018; Jurado et al, 2013). In our study, we also found that cLTP promotes significant changes in the proteome of cortical neurons. Pharmacological inhibition of DDHD2 and NMT1/2 clearly altered this response. The cLTP-induced protein expression responses, which were affected by IMP-1088 and KLH-45, displayed a high correlation, indicating a shared mechanism. Both DDHD2 and NMT1/2 inhibition impacted cLTP-induced pathways such as lipid metabolism, protein translation, and synaptic plasticity. Examples of this include: increased expression of sphingomyelin phosphodiesterase (Smpd1), a phospholipid modifying enzyme implicated in generation of lipid rafts enhancing NMDA receptor transmission (Wheeler et al, 2009) which has also been implicated as a potential marker for Alzheimer's disease (Florentinus-Mefailoski et al, 2021); ribosomal proteins (Mrpl55 and Rpl23) responsible for translation at the synapse (Dastidar and Nair, 2022); and phosphatidylinositol 3,4,5-trisphosphate-dependent Rac exchanger 1 protein (P-Rex1), a rho GTPase activator of the AKT signalling pathway (Ebi et al, 2013). Downregulated proteins of interest include: glypican 4 (Gpc4), a heparinated surface glycoprotein involved in stimulating the formation of excitatory glutamatergic synapses (Allen et al, 2012) and microtubule associated proteins 2 and 4 (Map2 and Map4), critical cytoskeleton scaffolding proteins involved in regulating neuronal structure and function (Kim et al, 2020; Tokuraku et al, 2010). Together, the patterns of dysregulated proteins explain the dramatic inhibition of the neuronal plasticity responses.

### N-terminal myristoylation is unaffected by cLTP

N-terminal myristoylation classically occurs co-translationally (Kosciuk and Lin, 2020), and was not affected in response to cLTP

or upon NMT1/2 inhibition in cultured cortical neurons. Myristoylated proteins have been reported to exhibit half-lives ranging from 2.3 to 21.2 days (Dörrbaum et al, 2018). Thus, it is not surprising that no change was observed in myristoylation over our 2 h experimental timeframes. Neither plasticity induction in vitro nor learning in vivo elicited a change in the prevalence of N-terminal modification, making it an unlikely avenue for DDHD2-mediated release of myristic acid to affect plasticity or learning outcomes.

## Post-translational lysine myristoylation contributes to synaptic plasticity

We have shown that inhibition of DDHD2 and NMT1/2 both completely block long-term potentiation in rodent brain slices as well as the increase in dendritic spine number and morphological changes in response to cLTP, demonstrating that myristoylation is critical to establishing synaptic plasticity. The primary role of lipidation, such as myristoylation, is to drive the association of proteins with membranes. Pharmacological inhibition of NMT1/2 activity also had an immediate and profound impact on the cLTP-induced neuronal high-density membrane (plasma membrane, Golgi and mitochondria) proteome, also suggesting a key role for post-translational lysine myristoylation. However, it is unlikely that all of the affected proteins are directly myristoylated by NMT1/2. It is more likely that most of the proteins undergoing membrane enrichment are controlled by one or more tethering protein(s) reliant on post-translational myristoylation as a translocation/vesicular trafficking signal. In support of this, only a small percentage of proteins displaying perturbed membrane localisation upon IMP-1088 treatment contained an annotated lipid modification. The scarcity of annotated protein lipidations among the IMP-1088 sensitive proteins implies that either a small number of post-translational myristoylation events mediate the observed changes in membrane localisation observed upon IMP-1088 treatment, or that multiple localisation mechanisms are at play. Post-synaptic proteins intimately associated with plasticity displayed IMP-sensitive membrane association, implying that post-translational myristoylation plays an integral part in mediating synaptic plastic responses. This was most strongly evidenced by the decreased membrane localisation of excitatory glutamatergic receptors in IMP-1088-treated neurons, including the NMDA receptor GluN1 subunit. Vesicular trafficking of this receptor to and from the post-synaptic membrane is the functional hallmark of LTP, indicating that post-translational protein myristoylation plays a

hitherto unexplored mechanism in synaptic plasticity (Diering and Huganir, 2018; Jurado et al, 2013). It is possible that IMP-1088 treatment affects the trafficking of vesicles harbouring these transmembrane proteins via disruption of Rab proteins. Dominant negative mutants of both Rab5 and Rab8 reduce NMDAR exocytosis (Gu and Huganir, 2016). While Rab5 is normally associated with the endocytic pathway, it also plays a role in exocytosis in mast cells (Klein et al, 2017) and in response to membrane damage (Bittel and Jaiswal, 2023). Further, the function of Arf6, which mediates the trafficking of glutamate receptors to the post-synapse (Oku and Huganir, 2013), is regulated by GTP-dependent lysine myristoylation (Kosciuk et al, 2020). Our work therefore suggests that post-translational myristoylation promotes synaptic plasticity by controlling post-synaptic receptor enrichment via vesicular trafficking. Lysine myristoylation may also contribute to synaptic plasticity by regulating the activity of Protein kinase A (PKA). After activation by cAMP, A kinase anchor proteins (AKAP) play a key role in regulating PKA activity by localising the kinase to specific subcellular compartments. Most notably among these are AKAP5 and AKAP12 (Qasim and McConnell, 2020), the latter of which has been previously demonstrated to undergo lysine myristoylation (Bagchi et al, 2022). Additionally, de novo lysine myristoylation of the C-terminal domains of R-Ras-2 and K-Ras-4 was previously shown to regulate Erk signalling. Post-translational myristoylation could therefore contribute to synaptic plasticity by regulating vesicular trafficking and signalling pathways.

## Effect of DDHD2 knock-out on hippocampal learning induced protein expression

Our in vivo experiments show that the hippocampal proteome associated with memory formation during instrumental conditioning is dramatically affected by the genetic ablation of DDHD2, especially post-synaptic proteins involved in synaptic plasticity (Grant, 2018). Indeed, altered expression of post-synaptic glutamate receptors, critical for LTP, was observed in these mice. In addition, microtubule-associated proteins 2/4 (involved in synaptic cytoskeletal organisation (Chazeau et al, 2016)), and growth arrest protein 7 (GAS7) (involved in PI3K-activated dendritic spine formation (Khanal et al, 2023)) were downregulated in DDHD2$^{-/-}$ mice and upon pharmacological inhibition of DDHD2 and NMT1/2.

Our findings establish a novel role for DDHD2 and NMT1/2 in synaptic plasticity, demonstrating their involvement in cLTP-induced proteomic changes, dendritic spine remodelling, and LTP formation in hippocampal circuits. Our results demonstrate that the myristoyl CoA pool generated in response to cLTP is rapidly utilised to promote de novo myristoylation, which plays a previously unsuspected role in synaptic plasticity rather than serving as a static post-translational modification. Given the transient nature of myristoyl CoA production and its potential impact on membrane integrity, it is likely that swift regulation of this intermediate takes place to maintain synaptic homeostasis. Our work highlights the importance of lipid metabolism in neuronal plasticity, suggesting a broader role for myristoylation in neuro-physiological processes beyond its traditional co-translational function. The correlation between pharmacological inhibition of DDHD2 in vitro and genetic DDHD2 ablation in vivo indicates

that these mechanisms are conserved across these experimental models. The identification of commonly affected pathways in synaptic function and plasticity, such as cytoskeletal organisation, dendritic spine formation and signalling, further underscores the functional relevance of de novo K-myristoylation. Future studies should explore the precise spatiotemporal dynamics of myristoylation in learning and memory, and determine whether disruptions in this pathway contribute to neurodevelopmental or neurodegenerative disorders. Given that DDHD2 mutations are associated with complex neurological disorders (Schuurs-Hoeijmakers et al, 2012), it will be critical to investigate whether impaired myristoylation contributes to synaptic dysfunction in these conditions.

## Methods

### Reagents and tools table

| Reagent/resource | Reference or source | Identifier or catalogue number |
| --- | --- | --- |
| **Experimental Models** | | |
| C57BL/6 | Ozgene | 000664 |
| DDHD2$^{-/-}$ | Scripps Research Institute Inloes et al, 2014 | |
| **Antibodies** | | |
| ATPB (3D5) | Abcam | ab14730 |
| Beta Actin (AC-15) | Abcam | ab6276 |
| EEA1 | Cell Signalling Technology | #2411 |
| GluN1 | Alomone Labs | AGC-001-GP |
| GM130 | Proteintech | 11308-1-AP |
| Legumain (D6S4H) | Cell Signalling Technology | #93627 |
| Map2 | Synaptic Systems | 188 004 |
| Mrpl55 | Thermo Fisher | PA5-103531 |
| PSD95 | Proteintech | 20665-1-AP |
| Sodium Potassium ATPase (ST0533) | Thermo Fisher | MA5-32184 |
| Synapsin1 | Synaptic Systems | 106 011 |
| IRDye 680RD Goat anti-Mouse IgG Secondary Antibody | Licor Bio | 926-68070 |
| IRDye 800CW Goat anti-Rabbit IgG Secondary Antibody | Licor Bio | 926-32211 |
| IRDye 800CW Donkey anti-Guinea Pig IgG Secondary Antibody | Licor Bio | 926-32411 |
| anti-guinea pig Alexa Fluor 647 | Thermo Fisher | A-21450 |
| anti-rabbit Alexa Fluor 546 | Thermo Fisher | A-11035 |
| **Chemicals, enzymes and other reagents** | | |
| 1,1-carbonydiimidiazole | Merck | 21860 |
| 2-amino-5-phosphonovaleric acid (APV) | Cayman Chemicals | 14540 |

| Reagent/resource | Reference or source | Identifier or catalogue number |
| --- | --- | --- |
| 2-mercaptoethanol | Merck | 63689 |
| 3-Pyridinemethanol | Merck | P66807 |
| Acetonitrile | Merck | 1.00029 |
| Ammonium formate | Merck | 156264 |
| Arachidic acid | Merck | A3631 |
| Arachidonic acid | Merck | 10931 |
| B-27 | Thermo Fisher | 17504001 |
| Bicuculline | Cayman Chemicals | 11727 |
| Chicken lysozyme | Merck | L6878 |
| Chloroform | Merck | 366927 |
| Cis-4,7,10,13,16,19-Docosahexaenoic acid | Merck | D2534 |
| Cis-5,8,11,14,17-Eicosapentaenoic acid | Merck | E2011 |
| Complete EDTA-free protease inhibitor cocktail | Roche | 11836170001 |
| Decanoic acid | Merck | C1875 |
| Dithiothreitol | Merck | D5545 |
| Docosanoic acid | Merck | 216941 |
| Dodecanoic acid | Merck | L4250 |
| Erucic acid | Merck | E3385 |
| Formic acid | Merck | 695076 |
| Glutamax | Thermo Fisher | 35050061 |
| Hexanoic acid | Merck | 153745 |
| Horse Serum | Invitrogen | 16050122 |
| IMP-1088 | Cayman Chemicals | 25366 |
| Iodoacetamide | Merck | A3221 |
| Iodoethane | Merck | I7780 |
| Iodoethane-d5 | Merck | 324582 |
| Iodomethane | Merck | 67692 |
| Iodomethane-d3 | Merck | 176036 |
| Iodopropane | Merck | 148938 |
| Isopropanol | Merck | 1.02781 |
| KLH-45 | Cayman Chemicals | 19889 |
| Lignoceric acid | Merck | L6641 |
| Linoleic acid | Merck | L1376 |
| Linolenic acid | Merck | L2376 |
| Lipofectamine® 2000 | Invitrogen | 11668500 |
| Methanol | Merck | 1.06035 |
| Myristic acid | Merck | M3128 |
| Myristoyl CoA | Merck | 870714 P |
| Nervonic acid | Merck | N1514 |
| N-ethylmaleimide | Merck | E1271-5G |
| neurobasal medium | Invitrogen | 1103049 |
| NuPAGE LDS sample buffer | Invitrogen | NP0007 |
| NuPAGE MOPS SDS running buffer | Invitrogen | NP0001 |
| Octanoic acid | Merck | C2875 |
| Oleic acid | Merck | O1008 |
| Palmitic acid | Merck | P5585 |
| Palmitoleic acid | Merck | P9417 |
| Palmitoyl CoA | Merck | 870716 P |
| Penicillin-streptomycin | Thermo Fisher | 15140122 |
| Pentadecenoyl CoA | Merck | 870715 P |
| Poly-L-Lysine | Merck | P2636 |
| ProLong Gold antifade | Invitrogen | P36934 |
| Sodium dodecyl sulfate | Merck | 71725 |
| SOLAµ HRP | Thermo Fisher | 60309-001 |
| Stearic acid | Merck | S4751 |
| Stearoyl CoA | Merck | 870718 P |
| Strychnine | Toronto Research Chemicals | TRC-S687713 |
| TBS blocking buffer | Licor Bio | 927-60001 |
| Tetrodotoxin | Cayman Chemicals | 14964 |
| Triethylamine | Merck | 90335 |
| Trifluoroacetic acid | Merck | T6508 |
| Trypsin (Proteomics grade) | Merck | T6567 |
| **Software** | | |
| cellSens | Olympus | |
| ImageJ | Fiji | |
| DIA-NN | GitHub Demichev et al, 2020 | |
| Protein Pilot | AB Sciex | |
| Graphpad Prism | | |
| In-house scripts Python | This study | |
| MultiQuant | AB Sciex | |
| Excel | Microsoft | |
| SPSS Statistics | IBM | |
| Cytoscape | Shannon et al, 2003 | |
| Image Studio | Licor Bio | |
| WinWCP | ohn Dempster, University of Strathclyde | |
| **Other** | | |
| 29 mm glass-bottom dishes with 10 mm micro-wells | Cellvis | #D29-10-1.5-N |
| Micro BCA Total Protein estimation kit | Pierce | 23235 |
| Immobilon-FL PVDF membranes | Millipore | IPFL85R |
| NuPAGE 4–12% Bis-Tris precast gels | Invitrogen | NP0321BOX |

## Ethical considerations and animals

For all experimental procedures, the care and use of animals was carried out in-line with the protocols approved by the Animal Ethics Committee of The University of Queensland (2017/AE000497, 2018/AE000508, 2021/AE000971, 2020/AE000352 and 2022/AE00073).

## In vivo experimental model and subject details

DDHD2$^{-/-}$ mice generated in a C57BL/6 background using standard gene targeting techniques (Inloes et al, 2014) were sourced from the Scripps Research Institute in the United States. The animals were maintained on a 12 h/12 h light/dark (LD) cycle at between 21 and 22 °C and housed in duos with access to standard mouse chow (in Dresden: Ssniff R/M-H; catalogue # V1534 and in Brisbane: Specialty Feeds, catalogue # SF00-100) and ad libitum autoclaved water. Brain extracts from instrumentally conditioned DDHD2$^{-/-}$ animals were sourced from our previous study, and the instrumental conditioning apparatus and behavioural procedures in that study are described in detail in (Akefe et al, 2024).

## In vitro neuronal culture

Cortical and hippocampal neurons were isolated from embryonic (E16) C57BL/6 mice. Cortical neurons were plated on poly-L-lysine coated six-well plates at a cell density of 500,000 neurons per well. Hippocampal neurons were seeded onto 29 mm glass-bottom dishes with 10 mm micro-wells (Cellvis, #D29-10-1.5-N) or 12 mm glass coverslips (ProSciTech) coated with 1 mg/mL poly-L-lysine at a density of 40,000 neurons per dish. Both cortical and hippocampal neurons were plated in neurobasal medium (Invitrogen) containing 5% horse serum media supplemented with 2% B-27 (Thermo Fisher), 2 mM Glutamax (Thermo Fisher) and 50 U/mL penicillin-streptomycin (Thermo Fisher). Media was exchanged for supplemented neurobasal media without horse serum after 2 h to allow for neuronal attachment to the plate. Neuronal cultures were grown at 37 °C under 5% CO$_2$ and fed twice per week with supplemented neurobasal media without horse serum.

## Glycine-induced chemical-LTP

Chemically-induced long-term potentiation (cLTP) studies were performed on cortical neurons at between 20 and 23 days in vitro (DIV) and on hippocampal neurons at DIV 21. This consisted of preconditioning the neurons for 1 h in artificial cerebrospinal fluid (ACSF) containing 125 mM NaCl; 2.5 mM KCl; 1.5 mM CaCl$_2$; 1 mM MgCl$_2$; 33 mM glucose buffered with 25 mM HEPES at pH 7.3 and the neurotoxin cocktail including 500 nM tetrodotoxin; 20 μM bicuculline; 1 μM strychnine. This was followed by 10 min stimulation using 200 μM glycine in ACSF neurotoxin formulation without MgCl$_2$. After stimulation, neurons were transferred to neurobasal medium supplemented with 2% B-27, 2 mM glutamax and neurotoxin cocktail. cLTP conditioned neurons were incubated for a further 30 min or 120 min post-glycine stimulation before subsequent sample processing. Negative control neurons were preconditioned as above for 30 min, after which they were transferred to ACSF with neurotoxins containing 50 μM 2-amino-5-phosphonovaleric acid (APV) for the remaining 30 min of preconditioning. Glycine treatment and the post-stimulation incubation were performed as above with the addition of 50 μM APV. Protein myristoylation inhibition experiments were performed as per LTP induction with the addition of N-myristoyl transferase inhibitor IMP-1088 (Caymen Chemicals) to final concentration of 500 nM. DDHD2 inhibition experiments involved preincubation of cultured neurons in neurobasal media containing 25 nM KLH-45 (Cayman Chemicals) for 3 h prior to cLTP treatment, followed by addition of 25 nM KLH-45 to the media for all subsequent steps.

## Dendritic spine analysis

Hippocampal neurons were transfected at DIV14 with the EGFP plasmid using Lipofectamine® 2000 (Invitrogen), as per the manufacturer's instructions. At DIV 21, neurons were subjected to cLTP and inhibitor treatment as described above. Two hours post-cLTP, neurons were fixed in 4% PFA-PBS for 20 min at room temperature. EGFP-expressing neurons were imaged using an Olympus UPLXAPO 60x/1.42 NA oil-immersion objective on a spinning disk confocal microscope (SpinSR10; Olympus, Japan) built around an Olympus IX3 body and equipped with two ORCA-Fusion BT sCMOS cameras (Hamamatsu Photonics K.K., Japan) and controlled by Olympus cellSens software. 3D Z-stack tiles images of neurons were acquired using the super resolution configuration, SoRa disk and the 3.2X magnifier. Primary, secondary and tertiary dendritic segments were analysed from each neuron. Dendritic spine number was manually quantified using ImageJ software and normalised to dendrite length. Spine morphology was analysed manually for three ROIs per cell, including spine head area, spine head width and spine length in ImageJ.

## GluN1 surface expression analysis

Mature hippocampal neurons (DIV 21) were pretreated with KLH-45, IMP-1088 or DMSO vehicle and subjected to cLTP as described above. 10 min post-cLTP neurons were fixed in 4% PFA-PBS for 20 min at room temperature, followed by incubation with GluN1 extracellular Ab (1/500 in PBS + 5% BSA) overnight. Cells were briefly washed (three times in PBS) and incubated with anti-guinea pig Alexa Fluor 647 (1/2000 in PBS + 5% BSA) for 2 h. Cells were washed again and permeabilised using Triton X100 0.01% in PBS for 5 min, washed and incubated with anti-PSD95 intracellular Ab (1/1000 in PBS + 5% BSA) overnight. Cells were washed, incubated with anti-rabbit Alexa Fluor 546 (1/2000 in PBS + 5% BSA) for 2 h, washed again and mounted in ProLong Gold antifade (Invitrogen). Coverslips were imaged using an Olympus UPLXAPO 60x/1.42 NA oil-immersion objective on a spinning disk confocal microscope (SpinSR10; Olympus, Japan) as described above. Multiple regions of interest were imaged as 3D Z-stacks using the super resolution configuration SoRa disk and the 3.2X magnifier. Analysis was performed in Imaris image analysis software (version 10.2.0). For each ROI, PSD95 and GluN1 were rendered as a surface using a consistent threshold between all conditions and filtered for volume < 1 μm$^2$. To solely analyse synaptic GluN1, this surface was further filtered using "shortest distance to surface [PSD95]" to select GluN1 within 0.5 μm of PSD95. Mean intensity for the GluN1 channel was extracted from statistics for the filtered surface.

## Membrane fractionation of cLTP-stimulated cortical neurons

Mouse cortical neuron cultures underwent treatment with either IMP-1088, KLH-45 or 0.1% v/v DMSO as a vehicle control coupled to cLTP as described in the previous section. Immediately after 10 min glycine stimulation, neuronal cultures were washed twice in ice-cold 10 mM PBS pH 7.4. Neurons were finally suspended in 200 µL hypoosmotic fraction buffer (10 mM KCl, 1 mM CaCl$_2$, 2 mM MgCl$_2$, 1 x Complete EDTA-free protease inhibitor cocktail (Roche), 1 mM dithiothreitol and 20 mM HEPES pH 7.3). Adherent neurons were then suspended using a cell scraper and transferred to a 1.5 mL centrifuge tube, followed by incubation for 30 min on ice to allow osmotic lysis. Lysates then underwent preliminary centrifugation at $720 \times g$ for 5 min at 4 °C in a benchtop microfuge (Beckman Coulter) to isolate the nuclear fraction (P1). The supernatant (S1) was then transferred to a fresh tube and centrifuged at $10,000 \times g$ for 5 min at 4 °C to isolate mitochondria (P2). The S2 supernatant was transferred to pre-chilled ultracentrifuge tubes and centrifuged at $188,000 \times g$ for 60 min at 4 °C in an Optima MAX-TL benchtop ultracentrifuge (Beckman Coulter). The purified high-density membrane fraction pellet (P3) immediately underwent proteomics or western blot analysis as described below. The protein content of supernatant (S3) was precipitated in 90% methanol at −20 °C overnight to isolate low-density membrane and cytosolic proteins. Precipitated proteins were pelleted by centrifugation in a benchtop microfuge at $15,000 \times g$ for 60 min at 4 °C. The pellet was resuspended in the appropriate lysis buffer for either proteomics or western blot analysis (see below).

## Synthesis of FFAST isotopic-coded differential tags

3-hydroxymethyl-1-methylpyridium iodide (FFAST-124), 3-hydro-xymethyl-1-methyl-d3-pyridium iodide (FFAST-127) and hydroxymethyl-1-ethylpyridium iodide (FFAST-138) tags were produced from commercially available CH3I/CD3I/C2H5I and 3-hydroxymethyl-pyridine reagents. The synthesis of FFAST derivatives was carried out as described previously (Narayana et al, 2015). In brief, 200 mg of iodomethane, iodomethane-D3, and iodoethane were each mixed with 100 mg of 3-hydroxy-methyl-pyridine. Under gaseous N$_2$, the resulting solution was heated to 90°C and microwaved for 90 min at 300 W in a CEM Discover microwave reactor. The resultant solution was then dried after being rinsed with 100% diethyl ether.

## FFA extraction and FFAST labelling of cLTP-treated neuronal cultures

In the cold room, frozen brain tissue samples were homogenised for 5 min in 0.5 mL of HCl (0.1 M) using a tissue homogeniser ultrasonic processor (Vibra-Cell, Sonics Inc., USA). About 200 mL of homogenate was treated with 0.6 mL ice-cold chloroform and 0.4 mL ice-cold methanol:12 N HCl (96:4 v/v, supplemented with 2 mM AlCl$_3$). After the mixture had been thoroughly vortexed, 0.2 mL ice-cold water was added, and the tubes were centrifuged at 4 °C for 2 min at $12000 \times g$ in a refrigerated microfuge (Eppendorf 5415 R). The upper phase was discarded, and the lower phase tube was dried in a vacuum concentrator (Genevac Ltd). Dried extracts were redissolved in 100 µL of acetonitrile. The derivatisation strategy was designed for molecules with free carboxylic acid

groups and followed a previously published procedure (Narayana et al, 2015). Briefly, 100 µL of FFA extracts in acetonitrile were combined with 50 µL of 1,1-carbonyldiimidiazole (1 mg/mL in acetonitrile) and incubated at room temperature (RT) for 2 min. Following this, 50 µL of either FFA extracts tagged with FFAST-124 or FFAST-127 (50 mg/mL in acetonitrile, 5% triethylamine) was added. The combinations were then mixed for 2 min before heating for 20 min at 50 °C in a water bath. Finally, 100 µL of each isotopically labelled sample was combined and dried in a vacuum concentrator. They were then redissolved in 200 µL of an internal standard solution (2.5 µM in acetonitrile) made by derivatising the 19 FFA standards with the FFAST-138 label. The samples were then placed in an autosampler vial and analysed using liquid chromatography tandem mass spectrometry (LC–MS/MS). Prior to analysis, all samples were maintained at −20 °C.

## FFA LC–MS/MS analysis

LC–MS/MS analysis was performed on a Shimadzu Nexera UHPLC equipped with a Poroshell 120 CS-C18, $2.1 \times 100$ mm column with 2.7 µm particle size, linked to an AB Sciex 5500 QTRAP tandem mass spectrometer fitted with an ESI Turbo V source. Analyst® 1.5.2 software (AB Sciex) was used for instrument control data collection, while Multiquant software (AB Sciex) was used for the multiple reaction monitoring (MRM) data analysis. Chromatographic separations were performed on 1 µL sample injection volume using a gradient system consisting of solvent A (0.1% formic acid (v/v)) and solvent B (100% acetonitrile with 0.1% formic acid (v/v)) was used to perform liquid chromatography (LC) at 0.450 mL/min heated to 60 °C. The gradient conditions consisted of an initial isocratic step of 15% B for 1 min, followed by a gradient to 100% B over 9 min. The column was flushed at 100% B for 2 min and then reduced back to 15% to re-equilibrate for 2 min for a total run time of 14 min. The first 0.5 min of the LC run was switched to waste to remove any excess underivatized FFAST tags.

Mass spectrometric data acquisition was performed using positive mode ionisation in MRM mode using the multiple reaction monitoring transitions described in Appendix Table S2. Ion source temperature was set at 400 °C, and ion spray voltage set to 5500 V. The source gases setting consisted of curtain gas, GS1 and GS2 set to 45, 40 and 50 psi, respectively. The collision energy, declustering potential and collision cell exit potentials were set at 50, 100 and 13 V, respectively, for all transitions.

## Acyl CoA extraction

Acyl CoA extraction was based upon previously published methods (ter Veld et al, 2009). After cLTP treatment, the media from each well was carefully aspirated and adherent neurons washed twice in 1 mL of ice-cold sterile 10 mM phosphate-buffered saline, pH 7.4, before final addition of 100 µL PBS. Adherent neurons were resuspended using tissue culture cell scrapper and cell suspensions transferred to 2 mL polypropylene LoBind safe-lock tubes (Eppendorf). Wells were washed in another 100 µL of ice-cold PBS and pooled with the cell suspension. Cellular material was pelleted using centrifugation for 10 min at $16,000 \times g$. The supernatant was removed, and the pellet resuspended in 500 µL of ice-cold extraction buffer (1:1:1 isopropanol/acetonitrile/10 mM ammonium acetate, pH 5). About 20 ng of pentadecenoyl CoA (C15:0

CoA) internal standard was added to each sample, followed by homogenisation of the neuronal lysates by repeated draw/elute cycles through a narrow-gauge needle. Cellular debris was pelleted by centrifugation at 4 °C for 10 min at 12,000 × g in a refrigerated benchtop centrifuge. The supernatant containing extracted acyl CoAs was transferred to a fresh tube. The cellular pellet was resuspended in another 500 μL of cold extraction buffer, recentrifuged, and supernatant pooled with the initial extract. Extracts were dried in a rotary vacuum evaporator at 50 °C and stored at −20 °C. Immediately before analysis, samples were reconstituted in 50 μL of 80% methanol.

## Acyl CoA LC–MS/MS

Quantification was performed on a Shimadzu 8050 triple quadrupole mass spectrometer (Shimadzu) fitted to an LC-30 Nexera HPLC (Shimadzu). Chromatographic conditions consisted of injecting 5 μL of neuronal acyl CoA extract onto a Kinetex C18 2.1 × 50 mm, 2.6 μm particle size column (Phenomenex) heated to 50 °C. Mobile phases were: A 10 mM ammonium bicarbonate pH 7.5, and B 95% acetonitrile with 10 mM ammonium bicarbonate pH 7.5, used at a flow rate of 0.4 mL/min. Separations were performed using a linear gradient from 5% B to 100% B over 6 min, followed by 3 min of column flushing and re-equilibration. Mass spectrometry acquisitions were performed using positive mode electrospray ionisation under the following source conditions; nebulising gas flow 3 L/min, heating gas flow 10 L/min, drying gas flow 10 L/min, interface temperature 300 °C, desolvation line temperature 250 °C, heat block temperature 400 °C. Multiple reaction monitoring (MRM) transitions for each target acyl CoA species were as follows Myristoyl CoA (C14:0 CoA) 1. 978.3 → 471.3 CE 38.4, 2. 978.3 → 428.0 CE 37.0; Palmitoyl CoA (C16:0 CoA) 1. 1006.4 → 499.4 CE 36.6, 2. 1006.4 → 428.0 CE 36; Stearoyl CoA (C18:0 CoA) 1. 1034.4 → 527.4 CE 35.4, 2. 1034.4 → 428.0 CE 37.8; Pentadecanoyl CoA Internal standard (C15:0 CoA) 1. 992.3 → 485.3 CE 36.0, 2. 992.3 → 428.0 CE 36 each with a 50 ms dwell time. All data was acquired and analysed using the Shimadzu Lab Solutions software.

## Proteomic sample processing

LTP induced neurons and membrane fractions immediately underwent protein extraction and processing for LC–MS/MS analysis. Neuron cultures, membrane fractions and brain samples were all processed identically. This consisted of suspending the sample in 200 μL of lysis buffer consisting of 50% v/v trifluoroacetic acid, 2% w/v sodium dodecyl sulfate, 10 mM dithiothreitol and 10 mM PBS buffer at pH 7.4, spiked with 1 μg of chicken lysozyme protein as an exogenous process control. Adherent neurons were suspended in the lysis buffer by scraping the bottoms of each well using a cell scraper. Lysates were then homogenised using ten draw and elute cycles through a 28-gauge needle, followed by centrifugation at 14,000 × g for 15 min in a benchtop microfuge to pelletize insoluble cellular debris. Supernatants were then heated to 65 °C for 45 min to denature proteins and reduce cysteine disulfide bonds. Free cysteines were then alkylated using 50 mM of iodoacetamide for 45 min at room temperature in the dark. Buffer exchange was performed by adding 1 mL of ice-cold methanol to lysates, followed by overnight incubation at −20 °C. Precipitated

proteins were pelletized by centrifugation at 14,000 × g for 15 min, and the pellet washed in 1 mL of ice-cold methanol. Samples for untargeted SWATH analysis were resuspended in 100 μL PBS at pH 7.4. Samples for protein lipidation analysis were resuspended in 80:20 PBS/acetonitrile to aid solubility of lipidated peptides. In addition to this 20 ng of a synthetic N-terminally myristoylated peptide Myr-GCVQCK modified with N-ethylmaleimide (NEM) was spiked into each sample as a process control. About 2 μg of proteomics-grade trypsin was added to each sample and digested into peptides overnight at 37 °C. Peptide digests were purified by passing the sample through SOLAμ HRP (Thermo Fisher) solid phase extraction cartridges using a positive pressure manifold. This consisted of washing cartridges in 200 μL of acetonitrile, priming with 200 μL of water, adding the sample to the cartridge and washing the bound sample twice in 200 μL of 0.1% v/v formic acid. Samples for SWATH analysis were eluted using 100 μL of 80% v/v acetonitrile, whereas peptide extracts for myristoylation analysis were eluted in 100 μL 1:1 acetonitrile/isopropanol. Extracted peptides were evaporated to dryness and stored at −20 °C. Immediately before LC–MS/MS analysis, extracted peptides for SWATH analysis were resuspended in 20 μL of 0.1% v/v formic acid, whereas samples for myristoylation analysis were resuspended in 20% acetonitrile/0.1% formic acid.

## High-resolution mass spectrometry: data-independent acquisition proteomics analysis

Quantitative untargeted proteomic analysis of neuronal peptide digests was performed on a 5600 TripleTOF mass spectrometer (AB Sciex) with a microflow LC (Eksigent). Chromatographic conditions consisted of injecting 2 μL of sample onto a 0.3 × 10 mm C18 micro trapping column (Phenomenex) heated to 40 °C under an isocratic flow of 0.1 mM formic acid at 5 μL/min. After 10 min, the trapping column was then switched in-line with the separation column (CL120 0.3 × 150 mm C18 with 3 μm particle size, Eksigent). Gradient mobile phases consisted of A 0.1 mM formic acid and B acetonitrile containing 0.1% formic acid, at a flow rate of 5 μL/min. Chromatographic separations started at 5% B at progressing to 32% B at 68 min, 40% B at 72 min and 95% B at 76 min, plateauing until 79 min, before dropping to 3% B at 80 min, followed by aqueous column re-equilibration for 7 min. Mass spectrometry acquisitions were performed using positive mode electrospray ionisation. Source conditions consisted of curtain, GS1, and GS2, gases set to 30, 30 and 20 psi, respectively, source temperature at 250 °C, and ion spray and declustering potentials set to 5500 and 100 V, respectively. Data were obtained using dataindependent SWATH acquisitions consisting of an initial MS1 survey scan covering a mass range between 350 m/z and 1800 m/z for 50 ms, followed by 42 MS2 acquisitions targeting a fixed 20 m/z precursor ion window between 400 m/z and 1250 m/z. Collision-induced dissociation (CID) for each of these ions was calculated using the rolling collision energy algorithm, scaling to ion mass. MS2 product ion scans covered a mass range between 200 m/z to 1800 m/z with an accumulation time of 80 ms each for a total cycle time 3.5 s.

Analysis of SWATH data were performed using DIA-NN software (Demichev et al, 2020) to identify and quantify proteins searched against the complete Mouse proteome obtained from the Uniprot Database (Consortium, 2021).

Induced protein expression threshold cut-offs were determined using exogenously spiked chicken lysozyme. As identical quantities of chicken lysozyme were spiked into all samples prior to initiating processing workflows, any variability in chicken lysozyme quantification provided an effective means of assessing the measurement uncertainty of the analytical method. In brief, the measured intensity values for all detected chicken lysozyme peptides in each sample were summed and normalised against the average summed intensity of all samples to give a chicken lysosome value for each sample relative to one. The fold change cut-off acceptance value was set at the mean ± 2 standard deviations observed in the normalised chicken lysozyme intensity values. Assuming normal distribution of the variance, this provides 95% confidence that any fold change between treatment conditions occurring at the cut-off value stems from sample biology rather than measurement variability. The highest standard deviation for chicken lysozyme observed was 13%, therefore the fold change acceptance cut-off was rounded to $\pm\log_2 0.3$ for the study.

## High-resolution mass spectrometry: myristoylation analysis

High-resolution mass spectrometry (HRMS) profiling of $N$-terminal myristoylation was performed as described for data-independent proteomic analysis with the following exceptions. About 4 µL of hydrophobic peptide extract was chromatographically separated using a linear gradient at 40 °C from 5% B to 95% B at 76 min, plateauing until 79 min, before dropping to 3% B at 80 min, followed by aqueous column re-equilibration for 7 min. MS acquisitions were performed using positive electrospray ionisation using information-dependent data acquisition (IDA) mode. IDA data acquisition was performed using an initial MS survey scan for 250 ms. From this, the top 30 most abundant ions with an intensity greater than 100 cps, a mass exceeding 350 Da and a charge state between 2 and 5 were selected for MS/MS CID fragmentation. CID collision energy for each of these ions was calculated using the rolling collision energy algorithm, scaling to ion mass. MS-MS spectra were acquired between 100 and 2000 $m/z$ for 55 ms.

Data analysis consisted of converting the IDA .wiff mass spectrometry data files to text-based mascot generic format (.mgf) file format and generation of .mgf files using Protein Pilot software. Information associated with the MS/MS myristoyl-glycine marker ion was performed using custom Python scripts. In brief, MS/MS spectra were scanned for the presence of the diagnostic 268.23 $m/z$ myristoyl glycine $b_1$ fragment ion with an intensity value above 20, with precursor ion intensity, $m/z$ value and retention time information extracted from positive spectra. The intensity values for precursor ions within 1 $m/z$ were summed to account for spectra derived from isotopic precursors. PCA was performed on the extracted marker ion data using the GraphPad Prism software platform. Two-dimensional cluster plots were generated using the dimensions with the greatest eigenvalues as calculated using the Prism PCA algorithm.

## N-terminal myristoyl Peptide LC–MS/MS

LC–MS analysis was performed on a Sciex 5500 QTRAP connected to a Shimadzu Nexera HPLC. Chromatographic conditions consisted of 2 µL injections onto a Phenomenex Aeris C18 column (2.1 × 100 mm, 2.6 µM particle size) heated to 60 °C. Mobile phases

consisted of A 0.1% formic acid and B 100% acetonitrile with 0.1% formic acid at a flow rate of 0.45 mL/min. Gradient conditions started at 15% B and went to 50% B over 15 min, followed by a sharp gradient to 100% B over 2 min. The column was washed at 100% B for 3 min followed 3 min re-equilibration at gradient starting conditions for a total run time of 23 min. Mass spectral data were acquired under positive mode ESI conditions using scheduled MRM conditions, where each transition was only monitored for a 1 min window around the designated retention time using a maximum duty cycle of 0.5 s. Source ionisation parameters were: Curtain, GS1 and GS2 gas flows of 30, 50 and 60 psi, respectively; source temperature was 550°C; and ion spray voltage 5500 V. Ion transmission voltages were: declustering potential 80 V, entrance potential 10 V and exit potential 9 V. The specific transition, CID collision energy and retention time conditions for each peptide target are listed in Appendix Table S1.

## Proteomic data analysis

Statistical analysis and visualisation of proteomic datasets was performed using Excel (Microsoft), Graphpad Prism and SPSS Statistics (IBM). Gene ontological analysis and determination of synapse- associated proteins was performed using the Panther Gene Ontology database (Thomas et al, 2022) and SynGO database (Koopmans et al, 2019). Functional and interaction network and pathway analysis was performed using the STRING (Szklarczyk et al, 2021) and Reactome (Gillespie et al, 2022) database and visualisation was performed using Cytoscape (Shannon et al, 2003).

## Western blotting

Samples for western blotting were suspended in detergent-based lysis buffer (150 mM NaCl, 0.5 mM EDTA, 1% v/v NP-40, 0.2% w/v sodium dodecyl sulfate, 10 mM Tris pH 7.4). Total protein content was determined using a Micro BCA total protein estimation kit (Pierce). About 40 µg of total protein from whole cell and hippocampal lysates or 15 µg for subcellular fractions was added to NuPAGE LDS sample buffer (Invitrogen) and 2-mercaptoethanol (Sigma) and heated at 95 °C for 5 min. Electrophoretic separation was performed using NuPAGE 4–12% Bis-Tris precast gels (Invitrogen) in NuPAGE MOPS SDS running buffer (Invitrogen) at 200 V. Proteins were transferred to Immobilon-FL PVDF membranes (Millipore) in Tris-Glycine transfer buffer (10 mM Tris, 100 mM Glycine, 20% v/v methanol, pH 8.6) at 100 V. Transfer times were dependent on the molecular weight of the target protein (45 min for proteins <50 kDa; 90 min for proteins >50 kDa). Membranes were blocked with intercept TBS blocking buffer (Li-cor) for 30 min prior to overnight incubation in primary antibody at 4 °C. Membranes were washed in TBS with 0.1% v/v Tween 20 before incubation in the secondary detection antibody for 1 h at room temperature. After washing in TBS with 0.1% v/v Tween 20, membranes were imaged in an Odyssey Fc (Li-cor) reader and analysed using Image Studio software (Li-cor).

## In vitro electrophysiological recordings

Control (C57BL/6) and DDHD2 knockout mice were deeply anaesthetised with 4% isoflurane and were transcardially perfused with an ice-cold oxygenated, sucrose-based artificial cerebrospinal

fluid (ACSF) containing (in mM): 248 sucrose, 3 KCl, 0.5 $CaCl_2$, 4 $MgCl_2$, 1.25 $NaH_2PO_4$, 26 $NaHCO_3$, and 1 glucose, saturated with 95% $O_2$ and 5% $CO_2$. Animals were decapitated, and the brain was removed and placed in ice-cold oxygenated sucrose-based ACSF cutting solution. 400 mm coronal slices were sectioned using a Leica VT1200S vibratome. Slices were then placed into room temperature ACSF containing (in mM): 124 NaCl, 3 KCl, 2 $CaCl_2$, 1 $MgCl_2$, 1.25 $NaH_2PO_4$, 26 $NaHCO_3$, and 10 glucose, saturated with 95% $O_2$ and 5% $CO_2$, pH 7.2–7.4, 305–315 mOsm. Slices containing dorsal hippocampus were selected for recording immediately (controls and knockouts) or were placed into oxygenated ACSF at room temperature containing either KLH-45 (25 nM) for 4 h or IMP-1088 (500 nM) for 1 h prior to recordings.

Slices were transferred to a chamber and continuously perfused with ACSF at 32 °C, whilst allowed to settle for 30 min. A tungsten bipolar stimulating electrode was placed into the Schaffer collaterals near the intersection of CA1 and CA2 in the dorsal hippocampus, and a recording electrode filled with ACSF placed into the stratum radiatum of CA1. Microelectrodes were pulled from borosilicate glass (1.5 mm outer diameter × 0.86 mm inner diameter) to a resistance of 1–3 MΩ using a vertical P10 pipette puller (Narishige). Stimulus strength was adjusted until an EPSP response was observed and then reduced to 50% strength (ensuring the response was still observed). Baseline responses were recorded for 20 min, following which a TBS stimulus was applied (15 bursts × 5 spikes at 100 Hz). Post-stimulus responses were then recorded for 60 min.

Electrophysiological signals were recorded using the I = 0 setting on a Multiclamp 700B Amplifier (Axon Instruments), low-pass filtered on-line at 10 Hz and acquired at a sampling rate of 25 kHz using an ITC-18-USB acquisition board (InstruTech) and WinWCP electrophysiology acquisition software (created by John Dempster, University of Strathclyde). Using the Easy Electrophysiology software, we determined LTP induction by quantifying the mean fEPSP slope during 10 min of baseline recordings and after LTP induction. Statistical analysis was performed in Graphpad Prism.

## Data availability

The data produced in this study is publicly available in the following databases:

Behavioural and quantified lipid abundance data: University of Queensland eSpace Data Collection (https://doi.org/10.48610/4a44503).

The mass spectrometry proteomics data have been deposited to the ProteomeXchange Consortium via the PRIDE (Perez-Riverol et al, 2022) partner repository with the dataset identifier (PXD061811) (https://www.ebi.ac.uk/pride/archive/projects/PXD061811).

Python scripts for quantitative and multivariate data analysis and visualisation are available upon request. Requests for software should be addressed to t.wallis@uq.edu.au (T.P.W.) and f.meunier@uq.edu.au (F.A.M.).

The source data of this paper are collected in the following database record: biostudies:S-SCDT-10_1038-S44318-025-00484-3.

## Peer review information

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

## Acknowledgements

This work was supported by an NHMRC Ideas Grant (2010901), awarded to F.A.M. and T.P.W., an NHMRC Senior Research Fellowship (1155794), an ARC LIEF grant (LE130100078) and a Clem Jones Centre for Ageing Dementia Research (CJCADR) Flagship grant awarded to F.A.M. This work was also supported by an NHMRC Ideas Grant (2027521) and a FWO Fundamental Research Grant (G057121N) awarded to N.D. The authors thank Elisabeth Mariott for technical expertise, the University of Queensland Centre for Clinical Research Mass Spectrometry Facility (UQCCR-MSF) for the provision of analytical services and technical advice. The University of Queensland Biological Resources Facility and Dr Rumelo Amor (QBI Advanced Imaging Facility) are also appreciated for the provision of support services and technical advice. Figures 1A, 2A, 4A and 5A were created with BioRender.com.

## Author contributions

**Benjamin Matthews**: Conceptualisation; Data curation; Formal analysis; Validation; Investigation; Visualisation; Methodology; Writing—original draft; Writing—review and editing. **Sevannah A Steeves**: Formal analysis; Investigation; Visualisation; Methodology. **Isaac O Akefe**: Data curation; Formal analysis; Investigation; Visualisation. **Noorya Yasmin Ahmed**: Formal analysis; Investigation; Visualisation. **Rachel S Gormal**: Investigation; Visualisation; Methodology. **Nathalie Dehorter**: Conceptualisation; Formal analysis; Funding acquisition; Visualisation; Writing—original draft. **Tristan P Wallis**: Conceptualisation; Software; Funding acquisition; Visualisation; Methodology; Writing—original draft; Writing—review and editing. **Frédéric A Meunier**: Conceptualisation; Resources; Supervision; Funding acquisition; Visualisation; Writing—original draft; Project administration; Writing—review and editing.

Source data underlying the figure panels in this paper may have individual authorship assigned. Where available, figure panel/source data authorship is listed in the following database record: biostudies:S-SCDT-10_1038-S44318-025-00484-3.

## Disclosure and competing interests statement

The authors declare no competing interests.

# Expanded View Figures

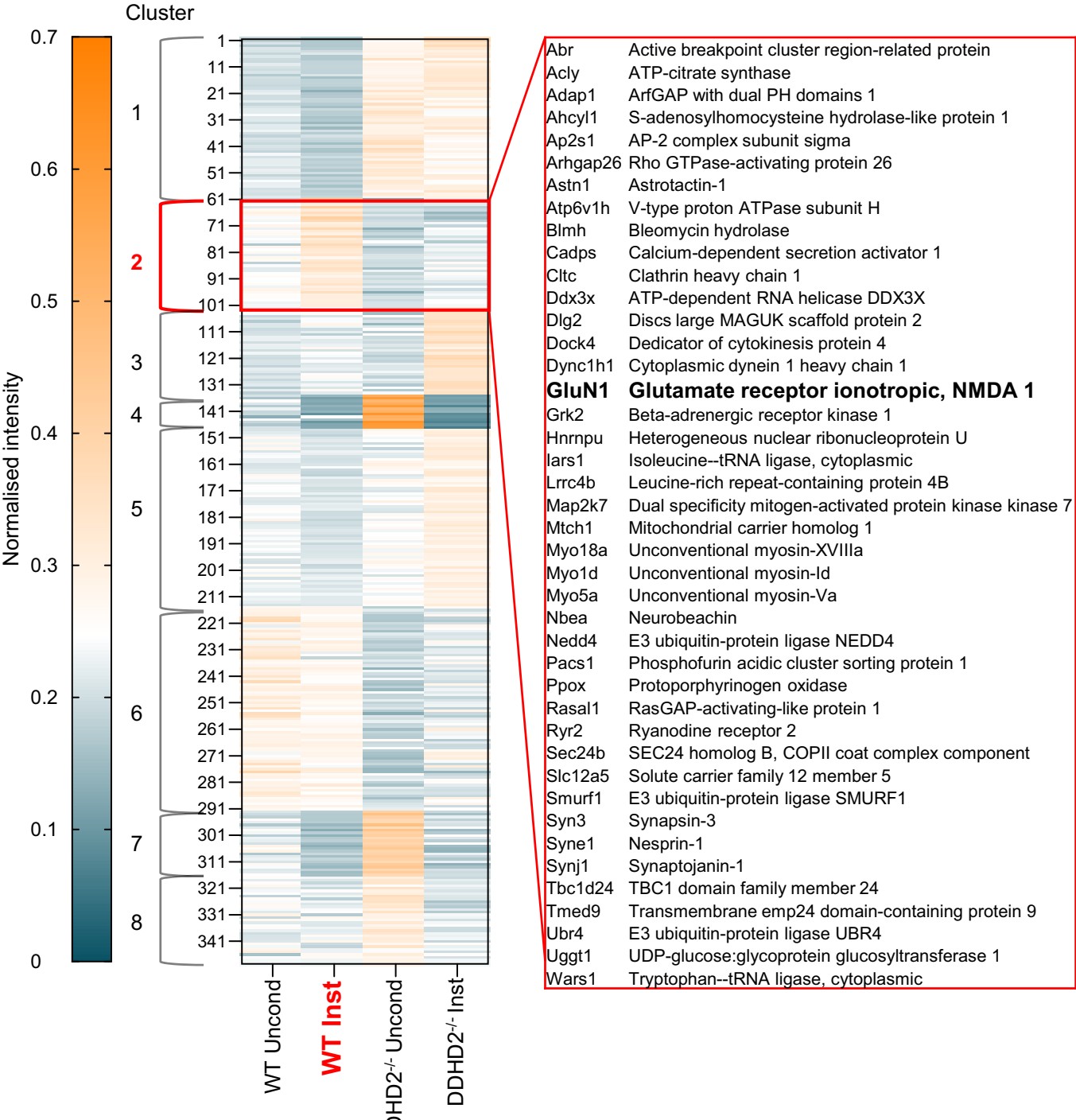

**Figure EV1.  Hierarchical clustered heatmap of the relative intensity of differentially expressed hippocampal proteins in instrumentally conditioned and control DDHD2$^{-/-}$ and C57BL/6 (wild type) animals.**

Cluster 2 proteins display learning responsive increased expression, which is ablated in DDHD2$^{-/-}$ animals. Data Information: Differential protein expression was determined by one-way ANOVA, with $p$ values $\leq 0.05$ considered significant. Relative protein intensity was calculated by dividing the average intensity value for each condition by the sum of the average intensities for all conditions. Hierarchical clustering of relative protein intensities was performed using Ward's method.

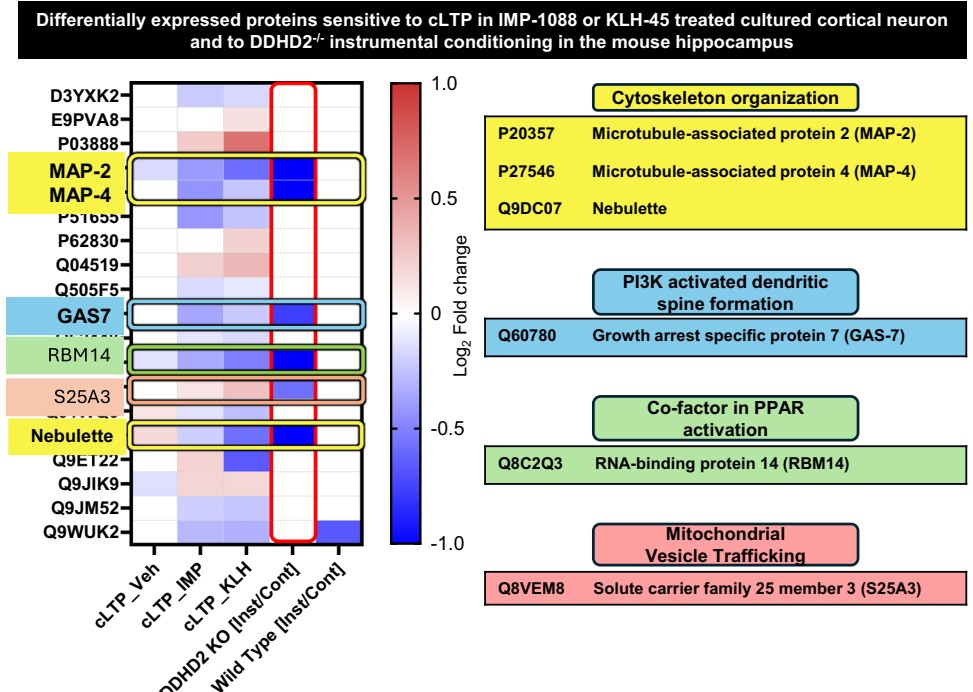

**Figure EV2. Differentially expressed proteins sensitive to cLTP in IMP-1088 or KLH-45-treated cortical neurons and to DDHD2−/− instrumental conditioning in the mouse hippocampus.**

Heatmap of the log₂ fold change in proteins significantly changed (*t*-test *p* values ≤ 0.05) 2 h post-cLTP in vehicle (DMSO) and inhibitor (KLH-45 or IMP-1088) treated cortical neurons in vitro and instrumental conditioning in the hippocampus of DDHD2−/− and C57BL/6 wild type 12-month-old mice.

