## [Peer Review File · The EMBO Journal]

Lysine myristoylation mediates long-term potentiation via membrane enrichment of synaptic plasticity effectors

Benjamin Matthews, Savannah Steeves, Isaac Akefe, Noorya Ahmed, Rachel Gormal, Nathalie Dehorter, Tristan Wallis, and Frederic Meunier

Corresponding author(s): Frederic Meunier (f.meunier@uq.edu.au) , Tristan Wallis (t.wallis@uq.edu.au)

Review Timeline:

Submission Date:	6th Jul 24
Editorial Decision:	20th Aug 24
Revision Received:	16th Mar 25
Editorial Decision:	16th Apr 25
Revision Received:	11th May 25
Accepted:	12th May 25

Editor: Ioannis Papaioannou

Transaction Report:

Dear Fred,

Thank you again for submitting your manuscript EMBOJ-2024-118403 for consideration by The EMBO Journal and for your patience during peer review. Your manuscript has been seen by three experts in the field, and we have received the full set of their comments, which I have already shared with you for your information (they are included again below). I would also like to thank you for your feedback on the referee reports and your tentative revision plan, which were helpful for us to reach a balanced and fair decision on your manuscript.

The referees recognize that the topic is relevant, and the findings potentially interesting and important for the field. However, they also identify several major limitations regarding technical aspects of the experiments, the reliance on omics data to draw conclusions, the inconclusive causality in the proposed link between protein myristoylation upon fatty acid generation and synaptic plasticity, and the limited extent of exploration of the proteomic datasets.

Upon discussion of the referees' input in our editorial team, and taking into account your expressed willingness to substantially revise your study by adding a significant amount of additional data to strengthen the manuscript and address the referee concerns, we are open to considering such a revised version of the manuscript. However, given the extent and nature of the referee concerns, the outcome of this process cannot be guaranteed and will depend on the completeness of your responses in the revised version and the referees' input in the next round.

Please include in your resubmission a detailed point-by-point response addressing all referees' comments. I should add that it is EMBO Journal policy to allow only a single round of major revision, and that we generally allow three months as standard revision time (November 19, 2024). As a matter of policy, competing manuscripts published during this period will not negatively impact our assessment of the conceptual advance presented by your study. However, we request that you contact us as soon as possible upon publication of any related work, to discuss how to proceed. Should you foresee a problem in meeting this three-month deadline, please let us know in advance and we may be able to grant an extension.

Thank you for the opportunity to consider your work for publication in The EMBO Journal. I look forward to your revision. Please let me know if you have any questions or comments that you would like to discuss with me in the meantime.

Best regards,

Ioannis

Instructions for preparing your revised manuscript

1. When you are ready to submit the revision, please upload:

- A Word file of the manuscript text (including legends of main Figures, EV Figures and Tables). Please make sure that changes are highlighted (or "tracked") to be clearly visible.

- Individual production-quality figure files (one file per figure). When assembling your figures, please refer to our figure preparation guidelines in order to ensure proper formatting and readability in print as well as on screen:

If the data shown in a figure are obtained from n {less than or equal to} 2, please use scatter plots showing the individual data points.

- i. the name of the statistical test used to generate error bars and P values
- ii. the number (n) of independent experiments (please specify technical or biological replicates) underlying each data point (discussion of statistical methodology can be reported in the Materials and Methods section, but figure legends should contain a basic description of n , P, and the test applied)
- iii. the nature of the bars and error bars (s.d., s.e.m.).

- A point-by-point response to the referees' comments, with a detailed description of the changes made (as a word file). All referees' concerns must be fully addressed and their suggestions taken on board. When preparing your letter of response to the referees' comments, please bear in mind that this will form part of the Review Process File and will therefore be available online to the community. Please note that you have the possibility to opt out of the transparent process at any stage prior to publication by letting the editorial office know (contact@embojournal.org); if you do opt out, the Review Process File link will point to the following statement: "No Peer Review File is available with this article, as the authors have chosen not to make the review process public in this case.". For more details on our Transparent Editorial Process, please visit our website: <https://www.embopress.org/page/journal/14602075/authorguide#transparentprocess>

- Expanded View (EV) files (replacing Supplementary Information) that are collapsible/expandable online. A maximum of 5 EV Figures can be typeset. EV Figures should be cited as "Figure EV1, Figure EV2" etc. in the text, and their respective legends should be included in the manuscript file after the legends of regular figures. See detailed instructions regarding Expanded View files here: <https://www.embopress.org/page/journal/14602075/authorguide#expandedview>

- For the figures that you do NOT wish to display as Expanded View figures, they should be bundled together with their legends in a single PDF file called "Appendix", which should start with a short Table of Contents (including page numbers). Appendix figures should be referred to in the main text as: "Appendix Figure S1, Appendix Figure S2" etc. Please see detailed instructions here: <https://www.embopress.org/page/journal/14602075/authorguide#expandedview>

- A complete author checklist, which you can download from our author guidelines (<https://www.embopress.org/page/journal/14602075/authorguide>). Please note that the checklist will also be part of the Review Process File.

2. Please note that no statistics should be calculated and shown in Figures if $n=2$. Please also note that each p value should be reported as an exact value.

3. Before submitting your revision, primary datasets (and computer code, where appropriate) produced in this study need to be deposited in appropriate public databases (see <https://www.embopress.org/page/journal/14602075/authorguide#dataavailability>).

In particular, you are kindly requested to deposit:

- i. the behavioural and quantified lipid abundance data,
- ii. all mass spectrometry proteomics data, and
- iii. any new software/code developed in this study

to appropriate public repositories. The accession numbers, databases, and the specific URLs (links) should be listed in a formal "Data availability" section (placed after Materials and Methods) that follows the model below (see also <https://www.embopress.org/page/journal/14602075/authorguide#dataavailability>):

Data availability

- RNA-seq data: Gene Expression Omnibus GSE46843 (<https://www.ncbi.nlm.nih.gov/geo/query/acc.cgi?acc=GSE46843>)
- [data type]: [name of the resource] [accession number/identifier/doi] ([URL or identifiers.org/DATABASE:ACCESSION])

*** All links should resolve to a page where the data can be accessed. ***

*** Please remember to provide in the Data availability section of your revised manuscript reviewer passwords if the datasets are not yet public. ***

*** The Data Availability Section is restricted to new primary data that are part of this study. In case you have no data that require deposition in a public database, please state so instead of referring to the database: "Our study includes no data deposited in public repositories." under the heading "Data availability". ***

*** Please use detailed data citations for already available datasets that were re-analyzed in your study - for more information on the format, see point #9 below. ***

4. Please check that the title and the abstract of the manuscript are brief, yet explicit, even to non-specialists. The length of the title should not exceed 100 characters, and the abstract should be a single paragraph not exceeding 175 words.

5. The Materials and Methods need to be described in the manuscript using our "Structured Methods" format, which is now required for all research articles. According to this format, the Materials and Methods section includes a single "Reagents and Tools Table" -listing key reagents, experimental models, software and relevant equipment and including their sources and

relevant identifiers- followed by a "Methods and Protocols" section describing the methods. More information on this format as well as detailed instructions, examples, and a template (.docx) for the "Reagents and Tools Table" can be found in our author guide: <https://www.embopress.org/page/journal/14602075/authorguide#structuredmethods>.

6. Please also note our reference format: <https://www.embopress.org/page/journal/14602075/authorguide#referencesformat>.

8. Please remember: digital image enhancement is acceptable practice, as long as it accurately represents the original data and conforms to community standards. If a figure has been subjected to significant electronic manipulation, this must be noted in the figure legend or in the "Materials and Methods" section. The editors reserve the right to request original versions of figures and the original images that were used to assemble the figure.

9. Our journal encourages inclusion of data citations in the reference list to directly cite datasets that were obtained from public databases. Data citations in the article text are distinct from normal bibliographical citations and should directly link to the database records from which the data can be accessed. In the main text, data citations are formatted as follows: "Data ref: Smith et al, 2001" or "Data ref: NCBI Sequence Read Archive PRJNA342805, 2017". In the Reference list, data citations must be labeled with "[DATASET]". A data reference must provide the database name, accession number/identifiers, and a resolvable link to the landing page from which the data can be accessed at the end of the reference. Further instructions are available at: <https://www.embopress.org/page/journal/14602075/authorguide#referencesformat>.

10. We request authors to consider both actual and perceived competing interests. Please review our policy (<https://www.embopress.org/page/journal/14602075/authorguide#conflictsofinterest>) and update your competing interests statement if necessary. Please name this section 'Disclosure and competing interests statement' and place it after the Acknowledgements section.

11. Please note that all corresponding authors are required to provide an ORCID ID upon submission of a revised manuscript (<https://orcid.org/>). Please find instructions on how to link your ORCID ID to your account in our manuscript tracking system in our Author guidelines (<https://www.embopress.org/page/journal/14602075/authorguide#authorshipguidelines>).

12. We use CRediT to specify the contributions of each author in the journal submission system. CRediT replaces the author contribution section, which should be removed from the manuscript. Please use the free text box to provide more detailed descriptions. See also guide to authors: <https://www.embopress.org/page/journal/14602075/authorguide#authorshipguidelines>.

14. We would also welcome the submission of cover suggestions or motifs to be used by our Graphics Illustrator in designing a cover.

15. Please use the link below to submit your revision:
<https://emboj.msubmit.net/cgi-bin/main.plex>

Referee #1:

The neuronal lipidome undergoes dynamic changes in response to neural activity and is potentially important for synaptic plasticity and learning and memory. Based on previous findings that the PLA1 enzyme DDHD2 generates saturated FFAs and in particular, myristic acid in response to neuronal stimulation, Matthews et al. hypothesized that myristic acid promotes synaptic plasticity. They show by LC-MS/MS analysis that DDHD2 generates sFFAs in response to chemical LTP, and myristic acid is rapidly converted to myristic CoA. They then hypothesized that myristic CoA is utilized for lipidation of synaptic proteins by NMT. By pharmacological inhibition of enzyme activities or using DDHD2 knockout mice, the authors determined the impact of the myristic acid metabolic pathway on synaptic proteins. By proteomic analysis they show that DDHD2 and NMT regulate the expression of a common set of synaptic proteins during cLTP. Importantly, they found that N-myristoylated proteins are stable and largely unaffected by LTP and learning, and activity-responsive release of myristic acid is not required for this form of post-translational modification. Analysis of the membrane proteome suggests that DDHD2 and NMT regulate membrane localization

of various synaptic proteins via lysine myristoylation. By electrophysiology analysis, the authors demonstrate that functionally, the DDHD2-NMT pathway regulates LTP.

Overall, this manuscript investigated the potential role of myristic acid in regulation of synaptic proteins via lipidation modification. The mechanistic roles of lipids in synaptic plasticity are important questions in the field of synaptic biology. The findings that membrane lipid remodeling provides a substrate for synaptic protein lipidation and regulates their membrane localization are novel. However, the membrane proteomic data are not technically convincing. Moreover, throughout the manuscript, the authors rely solely on the omics data to draw conclusions. Changes in both protein expression levels and membrane association need experimental validation to support the claim that the DDHD2-NMT pathway regulates membrane localization of synaptic proteins.

Major points:

1. Fig. 1D and E: why there was an increase in palmitoyl CoA in the control group (which was inhibited by KLH-45 treatment) in E but not in D after 10 min of cLTP induction?
2. DDHD2 inhibition abolishes the increase in palmitoyl CoA and stearoyl CoA, but only inhibits an increase in myristoyl CoA, raising the possibility that DDHD2 is not the only PLA enzyme responsible for the generation of myristic acid. The authors should discuss the contribution of DDHD2 and alternative metabolic pathway(s) to the generation of myristoyl CoA.
3. Quantification of acyl CoA in Fig. 1: Many proteins are myristoylated and/or palmitoylated. Fig. 1D and E shows a greater increase in myristoyl CoA than palmitic and stearoyl CoA generated in response to cLTP. What is the molar ratio of myristic CoA to palmitic or stearoyl CoA?
4. Fig. 3E: It looks like a statistically significant difference in learning-induced changes was only detected with GluN1, in addition to differences in GluR2 levels of WT and DDHD2^{-/-} unconditioned animals. Please clarify the conclusion that GluR1-3 and GluN1 displayed significant alterations in their learning-induced expression in DDHD2^{-/-}. As AMPA receptors are the key determinant of NMDA receptor-mediated plasticity, it is necessary to show changes in their expression levels by western blotting.
5. Fig. 5: Synaptic plasticity is expressed mostly via changes in the plasma membrane and plasma membrane-associated proteins, e.g., neurotransmitter receptors, t-SNAREs, and postsynaptic density proteins. In this study, the authors used crude membrane fractions to analyze membrane proteome. However, crude membranes from high-speed centrifugation are mixtures of various organelles and vesicles, making it difficult to determine the impact of lysine myristoylation on plasticity-related membrane proteins. It would be more appropriate to analyze plasma membrane proteins for synaptic plasticity studies. The authors could try to isolate plasma membrane by either density gradients or streptavidin affinity purification after surface biotinylation of cultured neurons.
6. By analyzing cLTP-induced proteome, the authors found that DDHD2 and NMT inhibitor treatment disrupts LTP-responsive protein expression. However, although several proteins were mentioned in the discussion, no direct evidence for changes in protein levels was provided. Western blotting of 2 or 3 proteins should be done to validate the results from proteomic analysis.
7. Analysis of membrane proteins suggests that lysine myristoylation might regulate protein localization during synaptic plasticity. Similarly, the authors mentioned RhoA, SNARE proteins, Rab5a/23, Numb, AP2, glutamate receptors, etc. as examples of NMT-regulated membrane proteome. Again, the authors should validate the proteomic data with western blotting of some candidate proteins in soluble and membrane fractions of cell or tissue lysates.
8. In the discussion, the authors claimed "GluR1 showed decreased membrane association in response to IMP-1088 treatment". Like other GluR subunits and other glutamate receptors, GluR1 is a transmembrane ion channel, not cytosolic. The decreased membrane association is most likely not caused by subcellular redistribution but rather changes in expression levels. Therefore, the authors should be cautious interpreting the data about integral/transmembrane proteins. Moreover, to validate the proteomics data, the authors should do immunofluorescence staining of GluR1 in cultured neurons.
9. As the major conclusion of the manuscript is "de novo lysine myristoylation mediates synaptic plasticity and LTP through membrane association of synaptic proteins", to make the case, I would suggest the authors categorize the IMP-1088 sensitive proteins into peripheral, lipid-anchored and integral membrane proteins, and compare the levels of peripheral and lipid-anchored proteins, e.g., Rabs or synaptic proteins known to be lysine myristoylated, in cytosolic and membrane fractions before and after cLTP stimulation of cultured neurons, and determine the impact of KLH-45/IMP-1088 on their membrane association.

Minor points:

1. In Fig. 6B, the fEPSP slope of control brain slices shows a huge drop in amplitude from the initial spike to levels close to the baseline, which is not typical of LTP expression in wild-type animals.
2. There are quite a few typos or spelling errors in the text. e.g., targetting, LPT.

Referee #2:

The manuscript by Matthews et al entitled "De novo lysine myristoylation mediates synaptic plasticity and long-term potentiation through membrane association of synaptic proteins" addresses a relevant scientific question concerning the mechanistic connection between lipid signaling and synaptic plasticity.

General Comments:

While the authors present an interesting set of findings, they fall short in showing that it is indeed protein myristoylation upon acute fatty acid generation that indeed drives synaptic plasticity. For instance, as the authors suggest in the Discussion section,

"DDHD2-derived FFAs generated in response to cLTP could also directly act as lipid messengers", which is a very relevant alternative. The authors should demonstrate experimentally that indeed myristoylation, for instance of synaptic proteins, could be driving these effects. Although it could be technically challenging to demonstrate that, testing if upon LTP/learning induced myristoylation specific synaptic proteins are being driven to membrane association and/or manipulating specific deacylases could be relevant approaches.

Also, many proteomic datasets were generated but specific proteins signatures are scarcely explored. This should be improved as it is a missed opportunity. Also, these proteomic datasets should be available ideally for the community and summary tables shown as supplementary information.

Major Specific Comments:

Figure1 - Since here is shown the main idea that gives the basis for this manuscript, can the authors demonstrate with other cLTP protocols that this phenotype also occurs, and is not specific for glycine? What about other forms of plasticity such as LTD?

- Even though myristic, palmitic and stearic acids increase to similar degree they are not converted equally to acyl-CoA. Can the authors address what could be happening to the differential fractions that are not converted?

Figure2 - The PCA analysis does not appear to clearly separate the groups as the authors claim. Can the authors comment on the presence of outliers concerning the proteomic analyses?

- Can the authors clearly describe which proteins are up or downregulated in response to cLTP, and connect these findings with known literature? Even though this is addressed partially in the Discussion section, it would be important to report these results in each proteomic analysis.

- Also, it would be relevant to validate, for instance by WB, some key proteins to strength these observations upon these manipulations?

Figure3 - How do these alterations compare with the ones observed in Figure2? Are there any common protein hits?

- Can the authors characterize the hits they get and connect with the literature in the WT after learning, as previously suggested in Figure2? Which specific hits are relevant upon DDHD2 KO and are these the same upon learning when compared with WT?

- Again, WB validation of specific hits would also be relevant.

- Finally, since many proteins are impacting synaptic proteins, how are synapses and spine number and morphology affected?

Figure4 - What do the authors mean by "minor changes"? Can they characterize those changes and put them into context?

Figure5 - There are many mechanisms that could be leading to differential membrane association beyond myristoylation. Can the authors show different number of proteins that are de facto membrane associated (e.g.: transmembrane proteins that should be solely or heavily enriched in this fraction - what's their % compared to other ones that should be the ones increased by cLTP, which should be partially cytosolic, but become more enriched in the membrane fraction? Can the authors expand with clear examples?

- Based on the proteins that are identified that are differentially regulated by DDHD2 or NMT inhibition, which ones have lysines or are known to have lysines to be myristoylated compared with other mechanisms? This would indirectly address the specificity of this mechanisms in cLTP.

Figure 6 - While DDHD2 and NMT inhibition led to loss of LTP induction, ablation of DDHD2 even looks like the LTP protocol leads to LTD. Can the authors address this potential additional plasticity issues in DDHD2? How differentially could LTD be also impacted and do the proteomic signatures provide some clues to potential connections with LTD alterations beyond LTP?

- Can the authors provide validation at the protein level (e.g.: through WB), with specific protein or proteins, based on proteomic hits, that indeed similar mechanisms are occurring in slice LTP experiments?

Discussion - Many interesting observations are discussed regarding specific protein hits from the proteomic analyses. However, these should be a follow-up on an overall description in the Results section (as mentioned earlier in the comments).

- "small percentage of proteins displaying perturbed membrane localisation upon IMP-1088 treatment contained an annotated lipid modification" - This is indeed a relevant observation to be discussed, but this analysis should have been shown more clearly in the Results section.

Methods

- "25 nM KLH45 for four hours or 500 nM IMP1088 for one hour prior to recordings" - Can the authors explain why they chose these concentrations and durations of treatment?

- A statistical sub-section would be useful to have in this Methods section. Additionally, throughout the legends of the figures or supplementary figures some explanations are given concerning statistical analyses. This should be reviewed, since not always this is performed, or the statistical tests used are not appropriately employed (e.g.: in many situations where Student T-Test was used, this might not be the correct test to be employed).

Minor Specific Comments

- Potential typos: "LPT", "DDHD", "Caymen".

- "Brain extracts from instrumentally conditioned DDHD2^{-/-} animals were sourced from our previous study" - It would still be useful and informative to briefly explain in this manuscript.
- Figure 1 Legend - Data Information - C should be D, D should be E.
- Figure 6 Legend - "Mean evoked post-synaptic potential slope across 50 minutes" - can the authors justify why 50 minutes?
- Figure 6 Legend - Data Information - "K)" - should it be KO?

Referee #3:

The manuscript by Matthew et al investigates the cellular response to pharmacological and genetic manipulation of the myristoyl-CoA synthesis pathway. The study is based on previous findings from the lab showing that free fatty acids increase during memory acquisition in vivo through DDHD2 activity. Using lipidomics and proteomics on cultured WT cortical mouse neurons, proteomics on hippocampal samples from DDHD2 and WT mice, and hippocampal field potential recordings in WT and DDHD2 knockout mice, all in combination with inhibition of key enzymes, the authors find here that DDHD2-derived myristic acid is rapidly conjugated to CoA to form myristoyl-CoA which triggers lysine-myristoylation in synaptic proteins underpinning synaptic plasticity.

This study is interesting and important as it reveals a new role for synaptic protein myristoylation in synaptic plasticity. The manuscript is also well-written and the study is executed well, with appropriate controls. However, there are certain points that require clarification, and both the introduction and discussion could be improved, as outlined below.

- The introduction could be more streamlined, to focus more on the central question investigated here.
- The color code in Fig. 1B/C is hard to track. It may be useful to particularly highlight the FFAs that change significantly (14:0, 16:0, and 18:0) and keep the others in a duller color.
- The authors mention that sFFAs are generated by DDHD2 in response to cLTP in Fig. 1, and while this is true for palmitoyl-CoA and stearoyl-CoA, DDHD2 inhibition does not completely block generation of myristoyl-CoA. The statement should therefore be adjusted.
- The rationale for the two hour wait period after LTP before analysis for most figures versus the analysis of protein lysine-myristoylation directly after cLTP needs to be explained better. How can these two data points be compared?
- The proteomic analyses are presented nicely, but lack focus on individual players that may be important. E.g. in Fig. 4, NMDAR and AMPAR are highlighted, but in Fig. 5, broad categories are shown. Are similar proteins affected here? And where are the details about the differential and common effects between KLH-45 and IMP-1088 treatment?
- For the experiments performed in neuronally enriched cultures, can the authors comment on the presence of non-neuronal cells and their contribution to the results?
- The discussion is rather long and focuses partially on specific changes. It would be helpful to add a model figure summarizing the study findings, and to highlight some of the most prominent changes.
- It is confusing that syntaxin-1, a transmembrane protein, changes its membrane association. How can a decrease in membrane association be envisioned? Is more syntaxin-1 degraded? Does syntaxin-1 mislocalize to other membranes that are excluded during the subcellular fractionation step? The authors could include more data for the membrane fractionation protocol, to demonstrate successful fractionation of P1, S1, P2, and P3, and maybe probe for a transmembrane protein such as syntaxin-1 to see where it may end up.

Minor changes:

- Legend to Fig. 1C: The authors state "analysis was performed on 4 independent time points". It is unclear where in the graph this is shown.
- Legend to Fig. 1E: Data information lacks panel (E) description.
- Please define "inst" and "uncond" in Fig. 3.
- Discussion: "Of all the SNARE proteins, membrane association of complexins 1 and 2 was most profoundly affected upon IMP-1088 treatment, ..." Complexins are not SNARE proteins, but SNARE-associated proteins.

Reviewer Comments

Reviewers' comments in **blue** and replies in **black**.

Referee #1

The neuronal lipidome undergoes dynamic changes in response to neural activity and is potentially important for synaptic plasticity and learning and memory. Based on previous findings that the PLA1 enzyme DDHD2 generates saturated FFAs and in particular, myristic acid in response to neuronal stimulation, Matthews et al. hypothesized that myristic acid promotes synaptic plasticity. They show by LC-MS/MS analysis that DDHD2 generates sFFAs in response to chemical LTP, and myristic acid is rapidly converted to myristic CoA. They then hypothesized that myristic CoA is utilized for lipidation of synaptic proteins by NMT. By pharmacological inhibition of enzyme activities or using DDHD2 knockout mice, the authors determined the impact of the myristic acid metabolic pathway on synaptic proteins. By proteomic analysis they show that DDHD2 and NMT regulate the expression of a common set of synaptic proteins during cLTP. Importantly, they found that N-myristoylated proteins are stable and largely unaffected by LTP and learning, and activity-responsive release of myristic acid is not required for this form of post-translational modification. Analysis of the membrane proteome suggests that DDHD2 and NMT regulate membrane localization of various synaptic proteins via lysine myristoylation. By electrophysiology analysis, the authors demonstrate that functionally, the DDHD2-NMT pathway regulates LTP.

Overall, this manuscript investigated the potential role of myristic acid in regulation of synaptic proteins via lipidation modification. The mechanistic roles of lipids in synaptic plasticity are important questions in the field of synaptic biology. The findings that membrane lipid remodeling provides a substrate for synaptic protein lipidation and regulates their membrane localization are novel. However, the membrane proteomic data are not technically convincing. Moreover, throughout the manuscript, the authors rely solely on the omics data to draw conclusions. Changes in both protein expression levels and membrane association need experimental validation to support the claim that the DDHD2-NMT pathway regulates membrane localization of synaptic proteins.

We thank the review for this fair assessment of our manuscript.

Major points:

1. Fig. 1D and E: why there was an increase in palmitoyl CoA in the control group (which was inhibited by KLH-45 treatment) in E but not in D after 10 min of cLTP induction?

Response: We thank the reviewer for pointing this out. We have now checked our data and have included a correction for this inter batch variability.

2. DDHD2 inhibition abolishes the increase in palmitoyl coA and stearoyl CoA, but only inhibits an increase in myristoyl CoA, raising the possibility that DDHD2 is not the only PLA enzyme responsible for the generation of myristic acid. The authors should discuss the contribution of DDHD2 and alternative metabolic pathway(s) to the generation of myristoyl CoA.

Response: The reviewer is correct that other pathway could be at play and we have altered the result section accordingly: "Unlike the sFFA sensitivity to KLH-45 treatment, myristoyl

CoA displayed a strong, but not complete inhibition of cLTP-induced sFFA CoAs. This is likely due to other possible source of myristic acid production.”

3. Quantification of acyl CoA in Fig. 1: Many proteins are myristoylated and/or palmitoylated. Fig.1D and E shows a greater increase in myristoyl CoA than palmitic and stearoyl CoA generated in response to cLTP. What is the molar ratio of myristic CoA to palmitic or stearoyl CoA?

Response: We thank the reviewer for pointing this out. We have now included the molar ratio of these changes and have included this new result in Fig. 1. We have also amended the results: “Examination of the relative molar concentrations of sFFA CoA conjugates at rest (APV treated) showed that myristoyl CoA only represent 0.3% of the trio (myristoyl palmitoyl and stearoyl CoAs) while palmitoyl and stearoyl CoAs represent 52.2% and 47.5% respectively (Fig 1E). This ratio is changed upon cLTP induction especially for myristoyl CoA now representing 3.2%, whereas palmitic CoA and stearic CoA represent 40.1% and 56.7% respectively (Fig 1E). This suggests that ACS substrate specificity favours the generation of myristoyl CoA during cLTP.”

4. Fig. 3E: It looks like a statistically significant difference in learning-induced changes was only detected with GluN1, in addition to differences in GluR2 levels of WT and DDHD2-/- unconditioned animals. Please clarify the conclusion that GluR1-3 and GluN1 displayed significant alterations in their learning-induced expression in DDHD2-/-. As AMPA receptors are the key determinant of NMDA receptor-mediated plasticity, it is necessary to show changes in their expression levels by western blotting.

Response: To align with the reviewer comments, we have decided to focus on the GluN1 change elicited by instrumental conditioning which are prevented in DDHD2-/-. We have further confirmed the change in GluN1 expression levels by western blotting and included this data in the new Fig 3 panel G. The reviewer is correct about the lack of effect detected on AMPA receptor subunit. For clarity, we have removed these panels from the figure.

5. Fig. 5: Synaptic plasticity is expressed mostly via changes in the plasma membrane and plasma membrane-associated proteins, e.g., neurotransmitter receptors, t-SNAREs, and postsynaptic density proteins. In this study, the authors used crude membrane fractions to analyze membrane proteome. However, crude membranes from high-speed centrifugation are mixtures of various organelles and vesicles, making it difficult to determine the impact of lysine myristoylation on plasticity-related membrane proteins. It would be more appropriate to analyze plasma membrane proteins for synaptic plasticity studies. The authors could try to isolate plasma membrane by either density gradients or streptavidin affinity purification after surface biotinylation of cultured neurons.

Response: We thank the referee for this suggestion. We have tried various home-made and commercially available kits and used subcellular fraction markers to confirm their purity. Our original ultracentrifugation method was the cleanest and we have now included the result of our western blot analysis on our two low- and high-density fractions (Fig. 5C). Our high-density fraction included plasma membrane, Golgi and some mitochondrial proteins. The scarcity of cortical neuron material obtained did not allow for further separation via biotinylation and/or sucrose gradient. We included an additional statement in the result to acknowledge this potential shortcoming: “The scarcity of cortical neuron material obtained did not allow for further separation of our fractions.”

6. By analyzing cLTP-induced proteome, the authors found that DDHD2 and NMT inhibitor treatment disrupts LTP-responsive protein expression. However, although several proteins were mentioned in the discussion, no direct evidence for changes in protein levels was

provided. Western blotting of 2 or 3 proteins should be done to validate the results from proteomic analysis.

Response: We have now included all proteins that underwent cLTP-induced enrichment in the high-density fraction and were affected by DDHD2 and NMT1/2 inhibitors including their level of inhibition (Fig 5I). We have also included several new EV tables 1, 2, and 4, detailing a full list of the proteins undergoing significant change in response to conditional learning in wt and DDHD2^{-/-} mice hippocampi and in cLTP with or without inhibitors (whole cell lysate and fractions are included). We have also validated the proteomics data from Fig 2 (see additional panels E-G) and Fig 3 with western blotting analyses (see panel G).

7. Analysis of membrane proteins suggests that lysine myristoylation might regulate protein localization during synaptic plasticity. Similarly, the authors mentioned RhoA, SNARE proteins, Rab5a/23, Numb, AP2, glutamate receptors, etc. as examples of NMT-regulated membrane proteome. Again, the authors should validate the proteomic data with western blotting of some candidate proteins in soluble and membrane fractions of cell or tissue lysates.

Response: As stated above, our many attempts to perform western blotting analysis were ultimately unsuccessful because of the scarcity of initial material following membrane fractionation. We have therefore decided to tone down our claims and remove any mention of most of these proteins. We have instead continued to focus on the NMDA receptor subunit GluN1 because its enrichment was decreased by both KLH-45 and IMP-1080 in our high-density membrane fraction, presumably because of underlying trafficking events. In addition, we carried out an orthogonal analysis of GluN1 synaptic surface expression by IF using of GluN1 extracellular Ab and PSD95 intracellular Ab in neurons 10 min post cLTP. We confirmed that KLH-45 and IMP-1088 treatments reduce GluN1 surface expression following cLTP (see new Fig 5J).

8. In the discussion, the authors claimed "GluR1 showed decreased membrane association in response to IMP-1088 treatment". Like other GluR subunits and other glutamate receptors, GluR1 is a transmembrane ion channel, not cytosolic. The decreased membrane association is most likely not caused by subcellular redistribution but rather changes in expression levels. Therefore, the authors should be cautious interpreting the data about integral/transmembrane proteins. Moreover, to validate the proteomics data, the authors should do immunofluorescence staining of GluR1 in cultured neurons.

Response: The reviewer is correct. We have now changed the result and included "recruitment and enrichment" related to the subcellular fractionation. We have also separated transmembrane protein enrichment levels from cytosolic recruitment in EV Table 4. Further, in line with reviewer #2, we have refocussed on NMDA receptor subunit GluN1 changes and have performed immunocytochemistry showing equivalent changes in surface expression (Fig 5J).

9. As the major conclusion of the manuscript is "de novo lysine myristoylation mediates synaptic plasticity and LTP through membrane association of synaptic proteins", to make the case, I would suggest the authors categorize the IMP-1088 sensitive proteins into peripheral, lipid-anchored and integral membrane proteins, and compare the levels of peripheral and lipid-anchored proteins, e.g., Rabs or synaptic proteins known to be lysine myristoylated, in cytosolic and membrane fractions before and after cLTP stimulation of cultured neurons, and determine the impact of KLH-45/IMP-1088 on their membrane association.

Response: We thank the reviewer for this comment. We have now included Expanded View tables (EV 4a and 4b) of all proteins displaying differential enrichment in high-density membrane fractions including if they contain transmembrane domains or known lipid modifications.

Minor points:

1. In Fig. 6B, the fEPSP slope of control brain slices shows a huge drop in amplitude from the initial spike to levels close to the baseline, which is not typical of LTP expression in wild-type animals.

Response: As reported in previous work (Rodrigues *et al*, 2021; Quintanilla, *et al*. 2024), the initial depression phase is a transient response that precedes the lasting synaptic potentiation, and this entire sequence is considered part of the induction of LTP. The drop in the fEPSP slope following TBS (which can occur due to the rapid activation of synaptic and intrinsic mechanisms in response to the intense burst stimulation) is part of the natural dynamics of synaptic plasticity where the system is "resetting." This response is variable and range between 50-300% change in slope (Rodrigues *et al*, 2021; Quintanilla, *et al*. 2024). Once the system adjusts to the TBS-induced changes, the synaptic strength is potentiated over time, indicated by a sustained increase in the slope. Fig 6B indicates that fEPSP slope remains significantly higher than the baseline, which is the hallmark of LTP.

2. There are quite a few typos or spelling errors in the text. e.g., targetting, LPT.

Response: We have corrected all detected spelling errors and uniformly applied the American spelling where applicable (eg. targeting versus targetting).

Referee #2

The manuscript by Matthews *et al* entitled "De novo lysine myristoylation mediates synaptic plasticity and long-term potentiation through membrane association of synaptic proteins" addresses a relevant scientific question concerning the mechanistic connection between lipid signaling and synaptic plasticity.

General Comments:

While the authors present an interesting set of findings, they fall short in showing that it is indeed protein myristoylation upon acute fatty acid generation that indeed drives synaptic plasticity. For instance, as the authors suggest in the Discussion section, "DDHD2-derived FFAs generated in response to cLTP could also directly act as lipid messengers", which is a very relevant alternative. The authors should demonstrate experimentally that indeed myristoylation, for instance of synaptic proteins, could be driving these effects. Although it could be technically challenging to demonstrate that, testing if upon LTP/learning induced myristoylation specific synaptic proteins are being driven to membrane association and/or manipulating specific deacylases could be relevant approaches.

Also, many proteomic datasets were generated but specific proteins signatures are scarcely explored. This should be improved as it is a missed opportunity. Also, these proteomic

datasets should be available ideally for the community and summary tables shown as supplementary information.

Response: The reviewer is correct. We have tabulated proteins differentially expressed in proteomic datasets presented in the results as we have presented all proteins undergoing significant change in the proteomics datasets reported in the manuscript as Expanded View tables (EV 1,2, 4a and 4b). We have also uploaded the raw data mass spectrometry to the Pride Database (Accession Number: PXD061811).

Reviewers can access the data using the below details

Reviewer access details

Log in to the PRIDE website using the following details:

Project accession: PXD061811

Token: r6lLdUeIlkUn

Alternatively, reviewer can access the dataset by logging in to the PRIDE website using the following account details:

Username: reviewer_pxd061811@ebi.ac.uk

Password: 02UZpwCA1G8a

1. Figure1 - Since here is shown the main idea that gives the basis for this manuscript, can the authors demonstrate with other cLTP protocols that this phenotype also occurs, and is not specific for glycine? What about other forms of plasticity such as LTD? Argue...

Response: This is a fair point. We have previously demonstrated that depolarisation of cortical neurons and secretagogue stimulation of chromaffin cells similarly increase sFFAs. Further, we have also demonstrated previously that both fear and instrumental conditioning similarly elicited an increase in sFFAs (Narayana *et al.*, 2015 Cell Chem Biol; Wallis *et al.*, 2021. Nature Comms; Akefe *et al.* 2024 EMBO) albeit in distinct regions of the brain.

2. Even though myristic, palmitic and stearic acids increase to similar degree they are not converted equally to acyl-CoA. Can the authors address what could be happening to the differential fractions that are not converted?

Response: We are not sure of what happen to the non-converted sFFAs. Presumably some could be lost via diffusion out of the cells.

3. -Mass balance calculations to determine the amount of myristoyl CoA

-The relative change in the palmitic acid

Response: We thank the reviewer for pointing this out. We have now included the molar ratio of these changes and have included this new result in Fig. 1. We have also amended the results: "Examination of the relative molar concentrations of sFFA CoA conjugates at rest (APV treated) showed that myristoyl CoA only represent 0.3% of the trio (myristoyl palmitoyl and stearoyl CoAs) while palmitoyl and stearoyl CoAs represent 52.2% and 47.5% respectively (Fig 1E). This ratio is changed upon cLTP induction especially for myristoyl CoA now representing 3.2%, whereas palmitic CoA and stearic CoA represent 40.1% and 56.7% respectively (Fig 1E). This suggests that ACS substrate specificity favours the generation of myristoyl CoA during cLTP."

4. Figure2 - The PCA analysis does not appear to clearly separate the groups as the authors claim. Can the authors comment on the presence of outliers concerning the proteomic analyses?

Response: We initially used PCA analysis to determine whether the proteomes changed with treatment conditions. Although we did observe obvious trends between cLTP alone and the inhibitor treatments, the reviewer is quite right about the presence of a few outliers disrupting clear clustering of the data. In view of this, we opted to remove these panels and focus more on the significant changes detected.

5. - Can the authors clearly describe which proteins are up or downregulated in response to cLTP, and connect these findings with known literature? Even though this is addressed partially in the Discussion section, it would be important to report these results in each proteomic analysis.

Response: We have tabulated proteins differentially expressed in proteomic datasets presented in the results as EV table 1, including cLTP induced changes in the vehicle controls.

6. Also, it would be relevant to validate, for instance by WB, some key proteins to strengthen these observations upon these manipulations?

Response: Yes. We have now included western blotting validation of the changes induced by cLTP upon pharmacological blockade of DDHD2 and NMT1/2. These confirmed our proteomics results. We thank the reviewer for requesting this important validation.

7. Figure3 - How do these alterations compare with the ones observed in Figure 2? Are there any common protein hits?

Response: This is a great question. We have now carried out this additional analysis and have generated a new EV Fig 2. Both pharmacological and genetic inhibition of DDHD2 led to common activated/inhibited pathways with a number of proteins similarly affected including (but not limited to) Map2, Map4, Gas7 and RBM14 which are involved in (1) cytoskeleton organisation, (2) PI3K activated dendrite formation, and (3) co-factor in nuclear FFA receptor (PPAR) activation. We have included a description of this additional analysis in the result section.

8. - Can the authors characterize the hits they get and connect with the literature in the WT after learning, as previously suggested in Figure2? Which specific hits are relevant upon DDHD2 KO and are these the same upon learning when compared with WT?

Response: We have performed an additional analysis of all the GluN1 interactors that are involved in learning and memory. This is now included in Fig 3F. We have also included a number of citations as requested by the reviewer.

9. Again, WB validation of specific hits would also be relevant.

Response: As also requested by reviewer #1 we have performed western blotting analysis of GluN1 and confirmed its upregulation in response to learning, an effect ablated in DDHD2^{-/-} animals.

10. Finally, since many proteins are impacting synaptic proteins, how are synapses and spine number and morphology affected?

Response: This is another great question. We have now carried out this analysis and are happy to report that cLTP-induced change in spine morphology and number is blocked by

DDHD2 and NMT1/2 inhibition. This is now an entirely new Fig 6 in the paper and we thank reviewer for suggesting this critical experiment.

11. Figure4 - What do the authors mean by "minor changes"? Can they characterize those changes and put them into context?

Response: We have clarified our message in the result section which now reads: "Similar to the result observed upon cLTP induction in vitro, no PCA clustering of the myristoyl glycine marker (Fig 4D left) or change in N-terminal myristoyl modified domain (Fig 4D centre) were observed in the learners versus non-learners hippocampi. We only found a few myristoylated proteins (Vsh1, Mic19, Tusc2 and Rnf141) displaying differential learning responsive regulation in DDHD2^{-/-} animals out of the 67 detected in the dataset (Fig 4E left). However, these proteins were not sensitive to learning. Given the broad alterations to the hippocampal proteome observed in response to genetic ablation of DDHD2 (Fig 3), the minor changes observed here is unlikely to be due to N-terminal myristoylation status."

12. Figure5 - There are many mechanisms that could be leading to differential membrane association beyond myristoylation. Can the authors show different number of proteins that are de facto membrane associated (e.g.: transmembrane proteins that should be solely or heavily enriched in this fraction - what's their % compared to other ones that should be the ones increased by cLTP, which should be partially cytosolic, but become more enriched in the membrane fraction? Can the authors expand with clear examples?

Response: Our data shows that disruption of DDHD2 and NMT1/2 affects vesicle trafficking which with a knock-on effect on transmembrane proteins, such as the NMDA receptor GluN1 subunit (Fig 5K), in the high-density membrane fraction. This hypothesis is supported by literature evidence showing that lysine myristoylation directly mediates plasma membrane localisation of ARF6 (Nat Commun. 2020, 11:1067), which controls the plasma membrane levels of Glutamate receptors (J Neurosci, 2013; 33:12586). We have now generated an EV table 4a and 4b that specify transmembrane protein from recruited proteins that are enriched in response to cLTP and sensitive to DDHD2 and NMT1/2 inhibitors. We have now expanded on the enrichment of NMDA receptor subunit GluN1 upon cLTP and its sensitivity to IMP-1088 and to a lower extent to KLH-45. In addition, we carried out an orthogonal analysis of GluN1 synaptic surface expression by IF using of GluN1 extracellular Ab and PSD95 intracellular Ab in neurons 10 min post cLTP. We confirmed that KLH-45 and IMP-1088 treatments reduce GluN1 surface expression following cLTP (see new Fig 5J).

13. Based on the proteins that are identified that are differentially regulated by DDHD2 or NMT inhibition, which ones have lysines or are known to have lysines to be myristoylated compared with other mechanisms? This would indirectly address the specificity of this mechanisms in cLTP.

Response: Unfortunately, there are several reasons why we cannot yet identify myristoylated lysin residues"

- 1- Lysine is a highly ubiquitous amino acid residue.
- 2- it is also the most posttranslational modified (acylation, succinylation, ubiquitination, etc.)
- 3- Lysine myristoylation is currently poorly characterised with only 3 known proteins have currently been shown to have lysine myristoylation (TNF α , Arf6 and Ras).

Therefore, there is insufficient literature evidence to validate our results.

14. Figure 6 - While DDHD2 and NMT inhibition led to loss of LTP induction, ablation of DDHD2 even looks like the LTP protocol leads to LTD. Can the authors address this potential additional plasticity issues in DDHD2? How differentially could LTD be also impacted and do the proteomic signatures provide some clues to potential connections with LTD alterations beyond LTP?

Response: The reviewer is correct that DDHD2^{-/-} seems to lead to a reduction in slope. In DDHD2^{-/-}, the LTP protocol seems to have induced an LTD. However, it is hard to compare this to LTD as the latter relies on low stimulation frequency to be established. A form of spike-timing dependent plasticity (STDP), where the relative timing of presynaptic and postsynaptic spikes determines whether LTP or LTD could have occurred in the DDHD2 KO. The combination of different patterns of synaptic activity can in some cases result in depression, particularly if the timing favours a net decrease in synaptic strength. Other mechanisms may be at play: Both LTP and LTD are NMDA receptor-dependent, but the difference lies in the intracellular calcium concentration. More modest rise in calcium levels in the DDHD2 KO could activate signalling pathways that lead to LTD rather than LTP, potentially involving dephosphorylation of AMPA receptors and a reduction in synaptic strength. Future work will be required to investigate whether DDHD2 also control synaptic depression.

15. Can the authors provide validation at the protein level (e.g.: through WB), with specific protein or proteins, based on proteomic hits, that indeed similar mechanisms are occurring in slice LTP experiments?

Response: Overall, our data demonstrate that genetic ablation of DDHD2 prevents memory formation and alters the membrane enrichment of key proteins involved in synaptic plasticity such as GluN1 (see Fig 3G). Similarly, our cLTP experiments demonstrate that DDHD2 and NMT1/2 are critically involved in regulating synaptic plasticity (LTP) and vesicular trafficking. In particular, Map2 and Prex1 are affected by DDHD2 and NMT1/2 inhibitor treatments (see new Fig 2). Our slice experiments demonstrate that the generation of LTP is completely blocked by pharmacological and genetic inhibition of DDHD2 and NMT1/2 inhibition. Furthermore, we have now demonstrated that cLTP-induced dendritic spine formation is also blocked by KLH-45 and IMP-1088 (new Fig 6). Our data strongly suggest that establishment of LTP is tightly controlled by de novo myristoylation. Because of time constraint, we could not use the slices for biochemical analysis, but we have included several new panels as outlined above to address the reviewer requests.

16. Discussion - Many interesting observations are discussed regarding specific protein hits from the proteomic analyses. However, these should be a follow-up on an overall description in the Results section (as mentioned earlier in the comments).

Response: We thank the reviewer for this observation. We have expanded Figure 5 adding panels (5I) describing the proteins undergoing inhibitor-induced changes in membrane enrichment and (5K) focused on changes in GluN1. We have also included an Expanded View table (EV table 4) describing all proteins undergoing significant changes in high-density membrane enrichment. Secondly, we have tightened the scope of the discussion to focus on inhibitor-induced effects on post synaptic receptors:

“This was most strongly evidenced by the decreased membrane localisation of excitatory glutamatergic receptors in IMP-1088 treated neurons, including NMDA receptor GluN1 subunit.”

17. "small percentage of proteins displaying perturbed membrane localisation upon IMP-1088 treatment contained an annotated lipid modification" - This is indeed a relevant

observation to be discussed, but this analysis should have been shown more clearly in the Results section.

Response: We have now included an expanded view table (EV table 4) with all proteins undergoing a significant change in enrichment in the high-density membrane fraction. We have segregated cLTP- and inhibitor-induced altered proteins based on the presence of transmembrane domain or known lipid modification. We have also changed the discussion to highlight this point more clearly.

18. Methods: - "25 nM KLH45 for four hours or 500 nM IMP1088 for one hour prior to recordings" - Can the authors explain why they chose these concentrations and durations of treatment?

Response: KLH-45 and IMP1088 doses and timings were based on Inloes et al., PNAS (2014) and Kallemeijn et al., Cell Chemistry Biology (2019) respectively.

19. Methods - A statistical sub-section would be useful to have in this Methods section. Additionally, throughout the legends of the figures or supplementary figures some explanations are given concerning statistical analyses. This should be reviewed, since not always this is performed, or the statistical tests used are not appropriately employed (e.g.: in many situations where Student T-Test was used, this might not be the correct test to be employed).

Response: We have made the corrections accordingly and have changed from t-test to ANOVA when appropriate.

Referee #3

The manuscript by Matthew et al investigates the cellular response to pharmacological and genetic manipulation of the myristoyl-CoA synthesis pathway. The study is based on previous findings from the lab showing that free fatty acids increase during memory acquisition in vivo through DDHD2 activity. Using lipidomics and proteomics on cultured WT cortical mouse neurons, proteomics on hippocampal samples from DDHD2 and WT mice, and hippocampal field potential recordings in WT and DDHD2 knockout mice, all in combination with inhibition of key enzymes, the authors find here that DDHD2-derived myristic acid is rapidly conjugated to CoA to form myristoyl-CoA which triggers lysine-myristoylation in synaptic proteins underpinning synaptic plasticity.

This study is interesting and important as it reveals a new role for synaptic protein myristoylation in synaptic plasticity. The manuscript is also well-written and the study is executed well, with appropriate controls. However, there are certain points that require clarification, and both the introduction and discussion could be improved, as outlined below.

1. The introduction could be more streamlined, to focus more on the central question investigated here.

Response: We have now modified the introduction to streamline the focus on the potential role of K-myristoylation on synaptic plasticity.

2. The color code in Fig. 1B/C is hard to track. It may be useful to particularly highlight the FFAs that change significantly (14:0, 16:0, and 18:0) and keep the others in a duller color.

Response: We have made the requested change in the figure and have further generated an extended figure with the molar ratio of the sFFA vs uFFAs response to cLTP as well that the relative ratio of the myristic, palmitic and stearic acids. We thank the reviewer for requesting these clarifications.

3. The authors mention that sFFAs are generated by DDHD2 in response to cLTP in Fig. 1, and while this is true for palmitoyl-CoA and stearoyl-CoA, DDHD2 inhibition does not completely block generation of myristoyl-CoA. The statement should therefore be adjusted.

Response: We have now added a paragraph addressing this point: "Unlike the sFFA, where KLH-45 treatment completely ablated the stimulation-induced myristic acid response, myristoyl CoA still displayed a reduced cLTP-induced elevation in the presence of the inhibitor, likely due to the CoA conjugation to the resting state pool of myristic acid in the cell (Fig 1B)."

4. The rationale for the two hour wait period after LTP before analysis for most figures versus the analysis of protein lysine-myristoylation directly after cLTP needs to be explained better. How can these two data points be compared?

Response: We have now clarified this point: Result section for Fig 2 has been altered: "In order to allow sufficient time for responsive protein translation to occur, we carried out this analysis 120 min after cLTP with or without inhibitors." Result section for Fig 5 has also been altered: "By identifying the membrane proteome that is sensitive to DDHD2 and NMT1/2 inhibitors immediately after cLTP stimulation (10 min), we hoped to capture changes that mainly are dependent on post-translational lysine myristoylation."

We have not compared these two experimental results directly as they are both informative however at different time scale.

5. - The proteomic analyses are presented nicely, but lack focus on individual players that may be important. E.g. in Fig. 4, NMDAR and AMPAR are highlighted, but in Fig. 5, broad categories are shown. Are similar proteins affected here? And where are the details about the differential and common effects between KLH-45 and IMP-1088 treatment?

Response: This is a fair point also raised by reviewer #1. To align with the reviewer #1's comments, we shift the focus from AMPA to NMDA receptor as the GluN1 change elicited by instrumental conditioning are blocked in DDHD2^{-/-}. We have further confirmed the change in GluN1 expression levels by western blotting and included this data in the new Fig 3 panel G.

We have now performed a network analysis identifying protein interactors of GluN1 that are differentially expressed in DDHD2^{-/-} and involved in synaptic plasticity. Similarly for Fig 5, we have now included a new analysis showing proteins enriched at the plasma membrane which are sensitive to DDHD2 and NMT1/2 inhibition (Fig 5G). Importantly, we found that GluN1 is highly sensitive to inhibition of myristoylation but less so of DDHD2 (Fig 5I).

6. For the experiments performed in neuronally enriched cultures, can the authors comment on the presence of non-neuronal cells and their contribution to the results?

Response: This is a fair point. Glial cells are certainly critically involved in the maturation of synapse. DDHD2 has so far been detected in neurons and indeed in glial cells such as astrocytes and oligodendrocytes (Protein Atlas). We have opted to focus on synaptic function because of the focus on synaptic plasticity and memory. Future work will be needed to pinpoint the role of glial cells in this process. We have included a sentence in the discussion to cover this omission: "Catabolic metabolism from supporting glia may contribute to some of the myristoyl CoA usage."

7. The discussion is rather long and focuses partially on specific changes. It would be helpful to add a model figure summarizing the study findings, and to highlight some of the most prominent changes.

Response:

8. It is confusing that syntaxin-1, a transmembrane protein, changes its membrane association. How can a decrease in membrane association be envisioned? Is more syntaxin-1 degraded? Does syntaxin-1 mislocalize to other membranes that are excluded during the subcellular fractionation step? The authors could include more data for the membrane fractionation protocol, to demonstrate successful fractionation of P1, S1, P2, and P3, and maybe probe for a transmembrane protein such as syntaxin-1 to see where it may end up.

Response: We thank the reviewer for this comment. We have now clarified our claim and performed additional western blotting analysis on our fractions (specifically the P3 fraction which we now refer to as the high-density membrane fraction in the text and S3 supernatant which we now refer to as the low-density membrane fraction in the text). The results for these additional analyses are displayed in figure 5C, showing the high-density membrane is enriched in plasma membrane, Golgi and mitochondria, while the low-density membrane contained EEA1 and Synapsin-1, markers for early endosome and synaptic vesicles respectively. Our data clearly shows that disruption of DDHD2 and NMT1/2 affects vesicle trafficking which with a knock-on effect on transmembrane proteins, such as the NMDA receptor GluN1 subunit (Fig 5K), in the high-density membrane fraction. This hypothesis is supported by literature evidence showing that lysine myristoylation directly mediates plasma membrane localisation of ARF6 (Nat Commun. 2020, 11:1067), which controls the plasma membrane levels of Glutamate receptors (J Neurosci, 2013; 33:12586). We have now amended the discussion to reflect these results. We have removed mention of syntaxin-1 from the discussion as not directly involved in synaptic plasticity.

Minor Changes

9. Legend to Fig. 1C: The authors state "analysis was performed on 4 independent time points". It is unclear where in the graph this is shown.

Response: We are sorry for this typo to 3 time points as indicated in the figure.

10. Please define "inst" and "uncond" in Fig. 3.

Response: This has now been defined

11. Discussion: "Of all the SNARE proteins, membrane association of complexins 1 and 2 was most profoundly affected upon IMP-1088 treatment, ..." Complexins are not SNARE proteins, but SNARE-associated proteins.

Response: The reviewer is correct. As requested by reviewers #1 and #2, we have altered the focus of the discussion to glutamate receptor localisation controlled by lysine myristoylation and have removed mention of complexin.

Dear Fred,

Thank you again for submitting your revised manuscript (EMBOJ-2024-118403R) to The EMBO Journal for our consideration, and for your patience during peer review. The revised version of your manuscript has now been seen by the three original referees who had previously assessed the first version of the manuscript, and we have received their comments, which are included below.

As you will see, all referees appreciate the thorough revision and acknowledge that all initially raised concerns have been adequately addressed. Given their supportive comments and recommendations, I am very pleased to say that your manuscript has in principle been accepted for publication in The EMBO Journal. Congratulations on a very successful revision!

Before we can proceed with formal acceptance of the manuscript, there are a few minor remaining issues pointed out by referees #1 and #2 (regarding unclear sentences, wrong Figure callouts, and typos) that we kindly ask you to address in a final version of your manuscript.

From the editorial side, there also a few changes and corrections we need from you in the final version of the manuscript. Please include in your resubmission a cover letter detailing how the points below are addressed and all changes to the manuscript:

- There is a name discrepancy for one of the co-authors that must be corrected: "Sevannah A. Steeves" in the manuscript vs. "Sevannah A Ellis" in the author's profile in our manuscript tracking system. Please make sure that the name is correctly provided in the revised manuscript and that the author's profile in our system is also updated/corrected as necessary.
- The funding information provided in the Comments box could not be extracted by our production team, and therefore all funders should be added to the "More Funders" list: Clem Jones Centre for Ageing Dementia (CJCADR) Flagship grant and FWO Fundamental Research Grant (G057121N).
- Please note that no more than 5 keywords can be listed after the Abstract of the manuscript (you currently list 9).
- The reviewer access details can now be removed from the "Data Availability" statement of the manuscript. Please make sure that the database, dataset ID, and permanent link (URL) to the specific dataset are provided for each deposited dataset. All deposited data must be publicly available at the time of publication.
- We noticed that there are callouts for Fig. 4E and 7E in the manuscript, but no such panels exist in Figures 4 and 7. Please correct this as necessary.
- There are no callouts for Fig. 5J; please make sure that all Figure panels are called out in the revised manuscript.
- The manuscript ID must be included in the general information table (at the top) of your Author Checklist.
- Source file names, titles, legends, and manuscript callouts all need to be updated to Dataset EV1-EV2 instead of Appendix Tables S1 and S3; they should be uploaded individually as Dataset files with their legends in a separate tab/sheet of each Excel file.
- The Appendix file needs to be in PDF format; Appendix Tables S2, S4-S5 should be compiled in the Appendix PDF; the title page of the Appendix should contain "Appendix for" followed by the manuscript title, and a Table of Contents with the page numbers for the listed items.
- Please rename the heading "Materials and Methods" to "Methods".
- Materials and methods need to be described in the manuscript using our structured methods format, which is now required for all research articles. According to this format, the Methods section includes a single "Reagents and Tools Table" -listing key reagents, experimental models, software and relevant equipment including their sources and relevant identifiers- followed by a "Methods and Protocols" section describing the methods. Used antibodies (currently in a separate table) must also be included in the Reagents and Tools Table. Please download and fill our Reagents and Tools Table template (.docx), which you can find in our author guide: <https://www.embopress.org/page/journal/14602075/authorguide#structuredmethods>. When submitting your revised manuscript, please do not include the Reagents and Tools Table in the Methods section of the manuscript but instead upload it as a separate file choosing the file type "Reagent Table".
- Please note that EMBO press papers are accompanied online by:
 - A) a short (2 sentences) summary of the findings and their significance,
 - B) 2-5 short bullet points highlighting the key results, and
 - C) a synopsis image in .jpg or .png format that is exactly 550 pixels wide and 300-600 pixels high (the height is variable). Please

note that the text needs to be legible at the final size.

Please upload this information along with your revised manuscript (the text for A and B should be provided in a separate Word file).

- During our routine pre-acceptance checks, our data editors have raised the following queries regarding figures, data, and legends. Please make sure that all requests below are completely addressed in the final version of your manuscript:

1. Please provide the exact p values in the legends of Figures 1B, F; 2E-G; 3G, 5J, K; 6C-E.
2. Please indicate what */ **/ ***/ **** represents; if this represents p value(s), please specify the exact p value in the legends of Figures 7B, D.
3. Please note that information related to "n" is missing in the legends of Figures 1F, 4B-D.
4. Please note that the error bars are not defined in the legends of Figures 2E-G; 3G, 5J, K; 6C-F.

Please also note that as part of the EMBO publications' Transparent Editorial Process, The EMBO Journal publishes online a Peer Review File along with each accepted manuscript. This File will be published in conjunction with your paper and will include the referee reports, your point-by-point response and all pertinent correspondence relating to the manuscript. You can opt out of this by letting the editorial office know (contact@embojournal.org). If you do opt out, the Peer Review File link will point to the following statement: "No Peer Review File is available with this article, as the authors have chosen not to make the review process public in this case."

We look forward to seeing a final version of your manuscript as soon as possible. Please let us know if you have any questions and use this link to submit your revision: <https://emboj.msubmit.net/cgi-bin/main.plex>.

Best wishes,

Ioannis

Referee #1:

The authors have thoroughly addressed all my comments and concerns through dedicated experiments, careful discussion, and editorial revisions. While the manuscript has been significantly improved and is now suitable for publication, additional editing is recommended to enhance clarity and ensure the text is polished for readability.

Minor points:

1. "Unlike the sFFA sensitivity to KLH-45 treatment, myristoyl CoA displayed a strong, but not complete inhibition of cLTP-induced sFFA CoAs."

This sentence is incomprehensible. Please rewrite it.

2. Fig. 4D: "We only found a few myristoylated proteins displaying differential learning responsive regulation in DDHD2^{-/-} animals out of the 67 detected in the dataset (Fig 4E left). However, these proteins were not sensitive to learning." First, there is no 4E in this figure. Second, the writing is confusing. Please correct and rewrite.

3. "This increase was sensitive to NMT1/2 inhibition suggesting that de novo lysine myristoylation is a requirement for this cLTP-induced protein levels (Fig 5F)."
Should be Fig 5E?

4. "Similarly, DDHD2 inhibition altered the membrane recruitment response to cLTP suggesting that myristic acid generated during cLTP is being used to drive de novo myristoylation (Fig 5E)."
Should be Fig 5F?

5. "Examination of the proteins with similarly disrupted membrane enrichment upon inhibition of DDHD2 and NMT1/2 activity revealed key proteins necessary for synaptic plasticity including the NMDA receptor subunit GluN1 and VSP29 (Fig 5I)."

VSP29 should be VPS29?

6. Fig 5J is not mentioned in the main text. The pseudocolor for GluN1 should be green to make it easier to visually distinguish from PSD95 signals in red.

7. Fig 5K: why no WB showing GluN1 levels in the high density membrane fractions?

8. Fig 6G and H are not mentioned in the main text.

Referee #2:

The authors have reasonably addressed the concerns that were raised by this reviewer.

Small typos should still be corrected as previously suggested:

-DDHD vs DDHD2 (pag7), Caymen vs Cayman

Referee #3:

The authors have addressed all of my concerns adequately. Thank you.

All editorial and formatting issues were resolved by the authors.

Dear Fred,

Congratulations on an excellent study! I am very pleased to inform you that your manuscript has now been accepted for publication in The EMBO Journal. Thank you for your thorough revision and for addressing the referees' concerns and our requests for editorial changes.

If you have any questions, please do not hesitate to contact the Editorial Office. Thank you for your contribution to The EMBO Journal. Working with you has been a pleasure!

Best wishes,

Ioannis
